# Nucleocapsids of the Rift Valley fever virus ambisense S segment contain an exposed RNA element in the center that overlaps with the intergenic region

Lyudmila Shalamova[1], Patrick Barth[2,8], Matthew J. Pickin [1], Kiriaki Kouti[1], Benjamin Ott [3], Katharina Humpert[3,9], Stefan Janssen [4], Gema Lorenzo[5], Alejandro Brun[5], Alexander Goesmann [2], Torsten Hain[3], Roland K. Hartmann [6], Oliver Rossbach[7] & Friedemann Weber [1] ✉

Rift Valley fever virus (RVFV) is a mosquito-borne zoonotic pathogen. Its RNA genome consists of two negative-sense segments (L and M) with one gene each, and one ambisense segment (S) with two opposing genes separated by the noncoding "intergenic region" (IGR). These vRNAs and the complementary cRNAs are encapsidated by nucleoprotein (N). Using iCLIP2 (individual-nucleotide resolution UV crosslinking and immunoprecipitation) to map all N-vRNA and N-cRNA interactions, we detect N coverage along the L and M segments. However, the S segment vRNA and cRNA each contain approximately 100 non-encapsidated nucleotides stretching from the IGR into the 5'-adjacent reading frame. These exposed regions are RNase-sensitive and predicted to form stem-loop structures with the mRNA transcription termination motif positioned near the top. Moreover, optimal S segment transcription and replication requires the entire exposed region rather than only the IGR. Thus, the RVFV S segment contains a central, non-encapsidated RNA region with a functional role.

Rift Valley fever virus (RVFV; genus *Phlebovirus*, family *Phenuiviridae*, order *Bunyavirales*) is a mosquito-borne zoonotic pathogen endemic in Africa. RVFV regularly causes small outbreaks, but occasionally also major and devastating epidemics leading to the death of thousands of cattle or sheep, and hundreds of humans[1]. Infection of humans usually results in a self-limiting febrile illness, but 1–2% can develop severe symptoms like hemorrhagic fever, encephalitis, or retinitis with case fatality rates of up to 70% among hospitalized patients[1–4].

RVFV particles contain mostly negative-stranded genomic RNA (vRNA) that is divided into three segments termed L, M, and S (Fig. 1A). The vRNAs encode the RNA-dependent RNA polymerase (RdRP; L segment), the envelope polyprotein M (M segment), and the N (nucleocapsid) protein (S segment). Upon cell infection, the vRNA segments are individually transcribed into mRNAs (primary transcription), subsequently replicated into antigenomes (cRNAs), and the cRNAs are then replicated back into vRNAs. The promoters for

[1]Institute for Virology, FB10-Veterinary Medicine, Justus-Liebig University, Giessen, Germany. [2]Bioinformatics & Systems Biology, Justus-Liebig University, Giessen, Germany. [3]Institute for Medical Microbiology, FB11-Medicine, Justus-Liebig University, Giessen, Germany. [4]Algorithmic Bioinformatics, Justus-Liebig University, Giessen, Germany. [5]Centro de Investigación en Sanidad Animal (CISA-INIA/CSIC), Valdeolmos, Madrid, Spain. [6]Institute of Pharmaceutical Chemistry, Philipps-University Marburg, Marburg, Germany. [7]Institute for Biochemistry, FB 08-Biology and Chemistry, Justus-Liebig University, Giessen, Germany. [8]Present address: Cell Biology and Plant Biochemistry, University of Regensburg, Regensburg, Germany. [9]Present address: Institute of Molecular Oncology, Genomics Core Facility, Philipps-University, Marburg, Germany. ✉e-mail: friedemann.weber@vetmed.uni-giessen.de

**Fig. 1 | RVFV particle structure, coding strategy, and iCLIP2 overview.**
**A** Schematic representation of a Rift Valley fever virus (RVFV) particle containing the L, M, and S segments' nucleocapsids. Each RNA is encapsidated by the N protein (green) and associated with the L polymerase (gray). **B** Simplified representation of the coding strategy of the RVFV segments. The L and M segments are negative-sense (viral RNA; vRNA) and the S segment employs an ambisense coding strategy (vRNA and complementary RNA; cRNA). For each segment, coding (dark gray) and non-coding (light gray) regions are depicted, the corresponding mRNAs are indicated by arrows. **C** Outline of the iCLIP2 procedure.

transcription and replication are formed by the partially complementary sequences at the noncoding 5′ and 3′ ends of each segment, the so-called panhandle[5,6].

The L and M segments employ a single-gene coding strategy as it is the norm for segmented negative-strand RNA viruses (Fig. 1B, top). Yet, phleboviruses like RVFV have expanded their genetic space by placing an additional reading frame (for the nonstructural protein NSs) on the S segment which is transcribed from the cRNA. In this "ambi-sense strategy", the two reading frames on the vRNA and cRNA are separated by a noncoding "intergenic region" (IGR), defined as the sequence between the two opposed stop codons (Fig. 1B, bottom)[7]. The IGR contains the transcription termination motifs of the two reading frames and was discussed to fold into a stem-loop, although the evidence is weak[8].

Both vRNAs and cRNAs are encapsidated by viral N and associate with the L polymerase to form nucleocapsids (or ribonucleoprotein particles; RNPs)[5,6]. Supported by structural and electron microscopy data, the prevailing view on negative-strand RNA viruses is that the vRNAs and cRNAs are uniformly bound by N protein[9–14]. Nonetheless, for influenza A virus (FLUAV), nucleotide resolution mapping of nucleoprotein (NP)-RNA interactions by CLIP (UV crosslinking and immunoprecipitation) detected numerous vRNA regions with poor NP coverage[15,16], and RNA structure analyses revealed short RNA structures and interactions between the segments[17,18]. However, FLUAV nucleocapsids differ as the RNA is wrapped around the NP and hence the bases are positioned on the outside, whereas for other segmented negative-strand RNA viruses the RNA is buried inside the RNPs[6,19]. For RVFV, for example, crystal structures show that the N protein has a central RNA binding cleft that forms a continuous groove on the inner surface of adjacent N subunits, thereby fully encapsidating the RNA[9,13,14].

Thus, as the encapsidation mode of FLUAV is not representative, the question of how the genome of segmented negative-strand RNA viruses is bound along the nucleocapsid has not been addressed sufficiently. Therefore, we mapped all RVFV N-RNA interactions for each vRNA and cRNA segment at single-nucleotide resolution and in a statistically robust manner, using nucleocapsids derived either from isolated RVFV particles or from infected cells. Our results indicate regular and conserved patterns of N coverage for all three segments of vRNA and cRNA, but also regions of higher-than-average coverage. In addition, stretches of non-encapsidated RNA were detected in the center of the genomic (vS) and antigenomic (cS) S segments. These exposed regions overlap with the IGR, have the potential to form a defined secondary structure and support RVFV S segment transcription and replication. To our knowledge, our study represents the first genome-wide nucleotide-resolution encapsidation profile for any negative-strand RNA virus besides FLUAV, provides data on the N abundance along each of the three RVFV vRNA and cRNA segments both in virions and in cells, and defines a novel non-encapsidated and structure-prone regulatory region in the center of the S segment.

## Results

### Mapping of N-RNA interactions in nucleocapsids from RVFV particles

We employed iCLIP2 (individual-nucleotide resolution UV crosslinking and immunoprecipitation)[20,21] as a method to provide a genome-wide map of the viral RNA sequences that are bound by the N protein of RVFV (Fig. 1C). In a first step, RNA and proteins were covalently crosslinked by UV-C irradiation either as part of isolated virus particles or within infected cells, followed by sample lysis to release the nucleocapsids. Then, partial RNase I digestion yielded N-bound RNAs of varying length that were immunoprecipitated with antibodies against N, followed by ligation of adapter oligonucleotides to RNA 3′ ends and $^{32}$P-labeling of RNA 5′ ends. The immunoprecipitated N-RNA complexes were subsequently separated by gel electrophoresis and transferred to a nitrocellulose membrane. Due to the radioactive 5′

label, the N-RNA complexes could be visualized by autoradiography (Fig. S1A, B). The radioactive complexes were excised from the membrane at a size range of 57–230 kDa that is expected for N (approximately 27 kDa) and its hexamers[9] plus 20–60 kDa of RNA and adapter[20]. The bulk of protein was then removed by Proteinase K treatment, which retains at least one amino acid or a small peptide at the crosslink side. The eluted RNA was reverse-transcribed up to one nucleotide upstream of the crosslinking site, where the reverse transcriptase stalls, defining the crosslink site in the 3′-end of the cDNA. After RNA degradation, the cDNA was ligated to a second adapter, followed by cDNA library preparation, PCR amplification, and size selection. Finally, the cDNA library was sequenced with up to 25 million strand-specific reads per sample, and mapped to the genome and antigenome of RVFV.

In addition to the standard iCLIP2 protocol outlined above, we took additional measures to increase the robustness and validity of the results. Firstly, for the immunoprecipitation step, we used two different mouse monoclonal antibodies in parallel in order to minimize potential biases introduced by a particular epitope. Secondly, every experiment was performed three times independently, and thirdly, as mentioned above, the entire iCLIP2 method was not only applied to virions but also to RNPs that were isolated from cells at 5 h post-infection (when replication is ongoing but RNPs are not yet removed by packaging into particles).

## N-RNA interactions in nucleocapsids from RVFV particles and infected cells

The collective data derived from the experiments using two monoclonal antibodies targeting different epitopes, each represented by three biological replicates (R1 to R3), are illustrated in Fig. 2 for virions and in Fig. S2 for infected cells. According to the convention for negative-strand RNA viruses, genomic-sense sequences representing

vRNA are depicted from 3′ to 5′, and positive-sense sequences representing cRNA from 5′ to 3′. Peaks indicate nucleotides interacting with N, and the peak height represents the number of reads assigned to each nucleotide. These overviews suggest that the iCLIP2 profiles were well reproducible for virions as well as for infected cells. Indeed, heatmaps comparing the biological replicates within each epitope group (Figs. S3A and S4A) as well as between the two epitope groups (Figs. S3B and S4B) demonstrated the high similarity between all iCLIP2 replicates within the virion group, and also within the infected cells group. Therefore, we built averages of the replicates from all 6 datasets per genomic- and antigenomic-sense segment per iCLIP2 group (virions or infected cells) to illustrate RNA coverage by RVFV N, first by pooling the data for the individual epitopes, and then by merging the epitope data (Figs. S3C and S4C). The overview and side-by-side comparison of the results in Fig. 3 show that the N iCLIP2 patterns from virions and infected cells differ, especially with respect to virions exhibiting increased encapsidation of vRNA and cRNA at the 3′ ends. In our view, this latter observation cannot be explained by classical defective interfering (DI) genomes with internal deletions, as these would have both ends overrepresented. Another possibility would be copy-back DI genomes, DIs that differ from conventional DI genomes by having a 3′ terminus that is exactly complementary to the 5′ terminus. The 3′ terminus of copy-back DI genomes would map as 3′ terminus of the antigenome. Hence, copy-back DIs are expected to exhibit comparable amounts of genome 5′ and antigenome 3′ ends, which again is not the case. Further possible explanations could be that the higher encapsidation rates at the 3′ ends either serve as a replication start signal (in case the RNP is the template) or a replication stop signal (in case the RNP is the product). In any case, the nature and purpose of the overrepresented 3′ ends in virion genomes is currently unclear. The differences of the iCLIP2 profiles between virions and infected cells could be due to the completed encapsidation in the

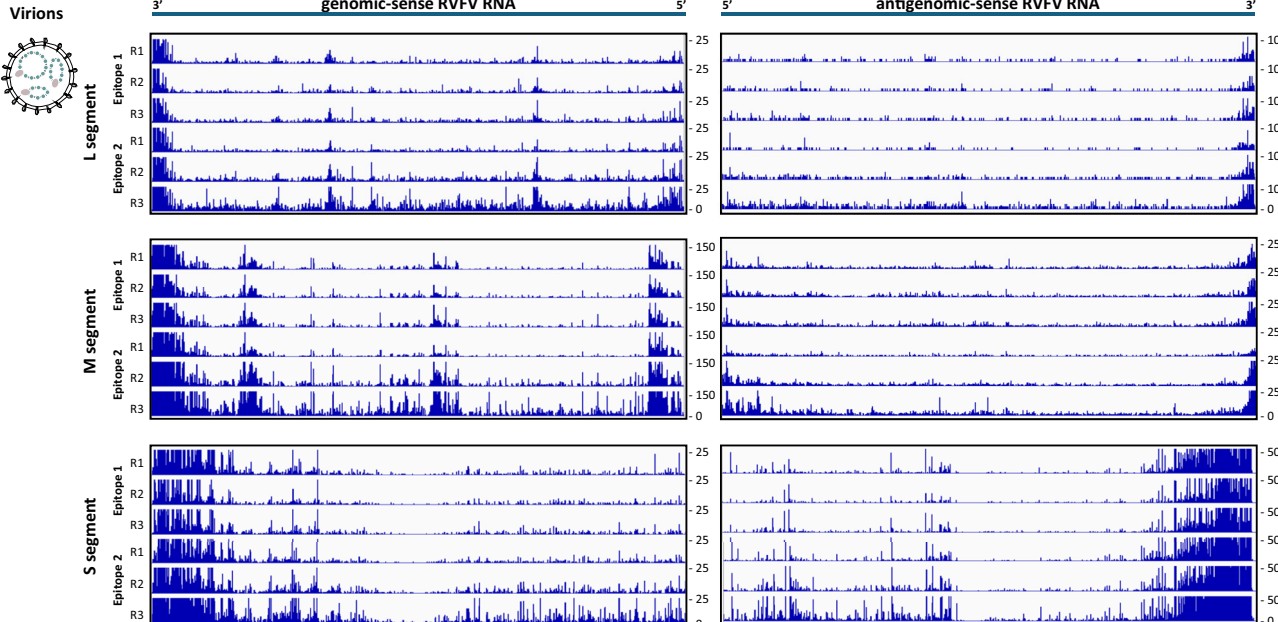

**Fig. 2 | Mapping of RVFV nucleoprotein-RNA interaction by iCLIP2 in virus particles.** Integrative Genomics Viewer (IGV) tracks representing the number of N-RNA crosslink reads (Y-axis) at single nucleotide positions (X-axis) along the RVFV MP-12 segment RNA sequences, as determined by N protein UV crosslinking and immunoprecipitation (iCLIP2) of virion ribonucleoprotein particles (RNPs). For each viral segment (L, M, or S, as schematically indicated on the left), viral RNA in genomic-sense orientation is shown 3′–5′ (vRNA) and 5′–3′ in antigenomic-sense orientation (cRNA). Two different monoclonal anti-N antibodies were used for

iCLIP2 experiments and data corresponding to each antibody are shown ("Epitope 1", "Epitope 2"), with three biological replicates in each case ("R1-R3"). Note that all panels were adjusted to the same width for visualization of good reproducibility of all six iCLIP2 patterns, although the segment lengths vary between 6404 (L), 3885 (M), and 1690 (S) nt. Track height range (crosslink site count) is adjusted by autoscale and presented on the right-hand side of each track. In this and subsequent iCLIP2 representations, reads from the 3′ and 5′ ends are shown out of range to obtain optimal resolution of the central parts of the sequences.

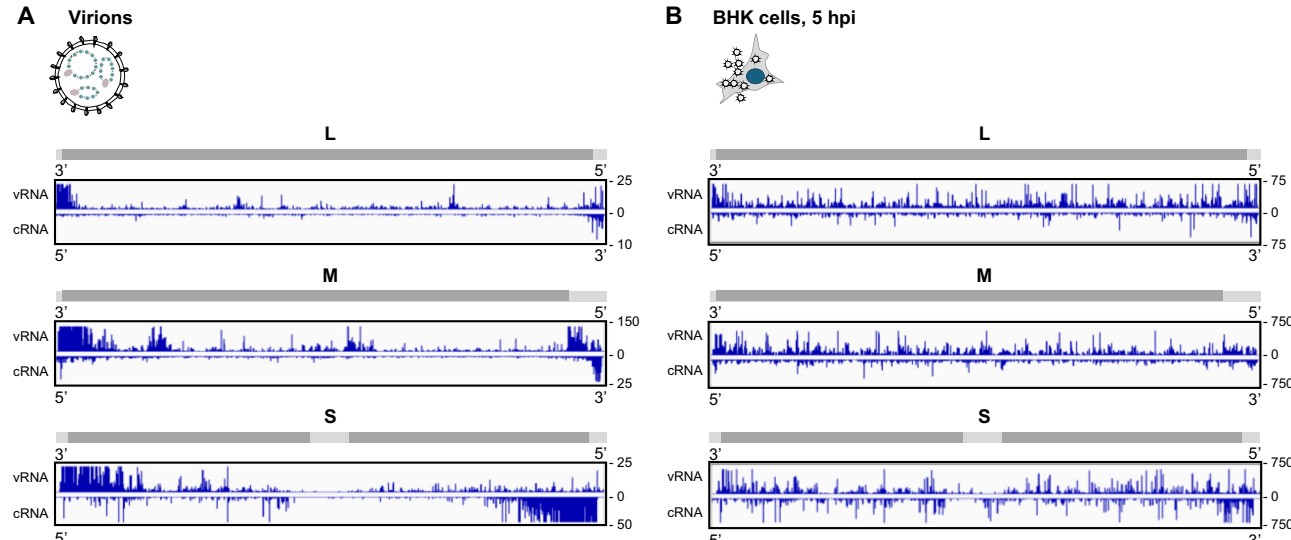

**Fig. 3 | RVFV nucleoprotein-RNA interaction in virions and in infected cells.** IGV tracks summarizing RVFV N binding along all genomic („vRNA", shown 3′–5′) and antigenomic segments („cRNA", shown 5′–3′) of MP-12 virions (**A**) and MP-12-infected BHK-21 cells (5 h p.i) (**B**). The tracks are the average of six biological replicates (three replicates per epitope) and grouped by RVFV segment (L, M, and S). For each segment coding (dark gray lines) and noncoding (light gray lines) regions are depicted. Track height range (crosslink site count) is shown on the right side of each track.

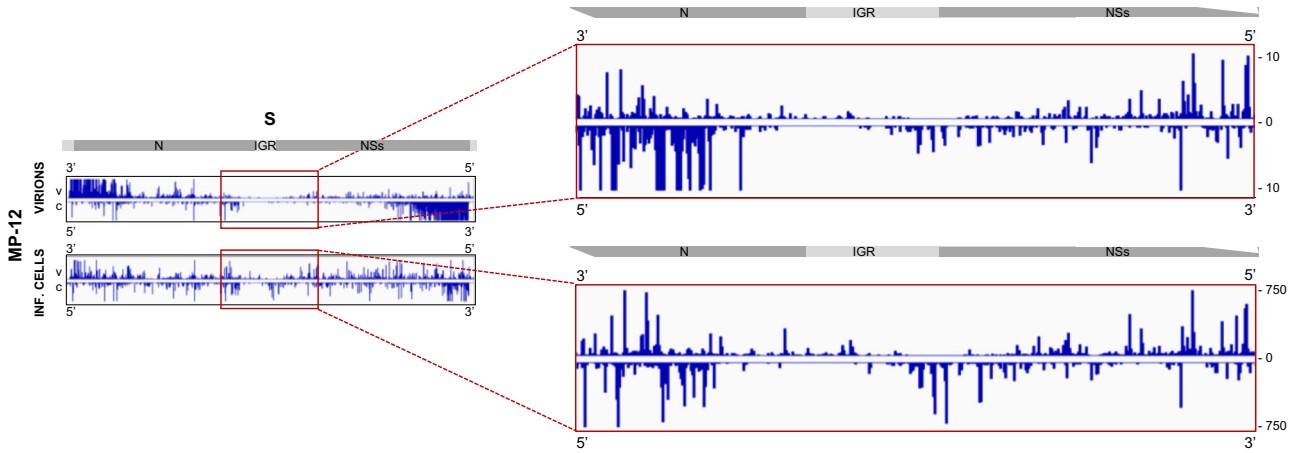

**Fig. 4 | RVFV N-iCLIP2 reveals low nucleoprotein coverage in the proximity of the intergenic region.** IGV tracks summarizing RVFV N association along the genomic (v) and antigenomic (c) sequence of the S segment in RVFV MP-12 virions or MP-12-infected BHK-21 cells (see Fig. 3). Close-up views of the intergenic region (IGR)-proximal region are shown on the right. The tracks are averages of six biological replicates in total (three replicates per epitope). Presentations of segment regions and IGV tracks are as for Fig. 3.

virions vs. the ongoing replication and hence different encapsidation stages in infected cells. Nonetheless, the segment-specific vRNA and the cRNA encapsidation patterns are consistent for both the virion-borne and the intracellular RVFV nucleocapsids. Moreover, encapsidation occurs primarily along the entire segment length, though not in a uniform manner.

### S segment nucleocapsids contain a non-encapsidated region

Our iCLIP2 data also show that both the genomic and antigenomic S segment nucleocapsids, but not those of the L or M segment, contain a region with conspicuously low N protein coverage (see Figs. 2 and S2). Interestingly, the location in the center of the segment is not identical with the IGR as it includes coding regions (Fig. 4). Moreover, there are differences between the genomic and the antigenomic data, as regions that are devoid of N on the genomic vS are encapsidated in the antigenomic cS and vice versa. However, the results of iCLIP method can be slightly biased as they require a broad size range of cDNAs[22].

Although we have accounted for this by using a read length of 150 nt, we tested the presence of exposed RNA regions with an independent method, namely limited RNase treatment. RVFV RNPs were isolated from concentrated virions, incubated for 3 min with increasing amounts of the ssRNA-specific RNase I, and then the RNA was extracted and subjected to northern blot analysis detecting the 3′ ends of vS or of cS (Fig. S5, thick black lines). As shown in Fig. 5A (left panel), incubation at the two highest RNase I concentrations reduced the size of RNP-borne full-length vS RNA (1690 nt) to clear-cut bands of ~900 nt. When naked RNA isolated from virions was used instead of RNPs, RNase I treatment did not produce a distinct product band and eventually resulted in complete degradation at the highest RNase I concentration (Fig. 5A, right panel). Also in the case of the antigenomic cS RNA, RNase I generated a shortened, yet more diffuse band with RNPs as substrate, but a smear and eventual complete degradation with isolated RNA as substrate (Fig. 5B). In contrast, limited RNase I digestion of M segment RNPs did not yield any distinct RNA bands at

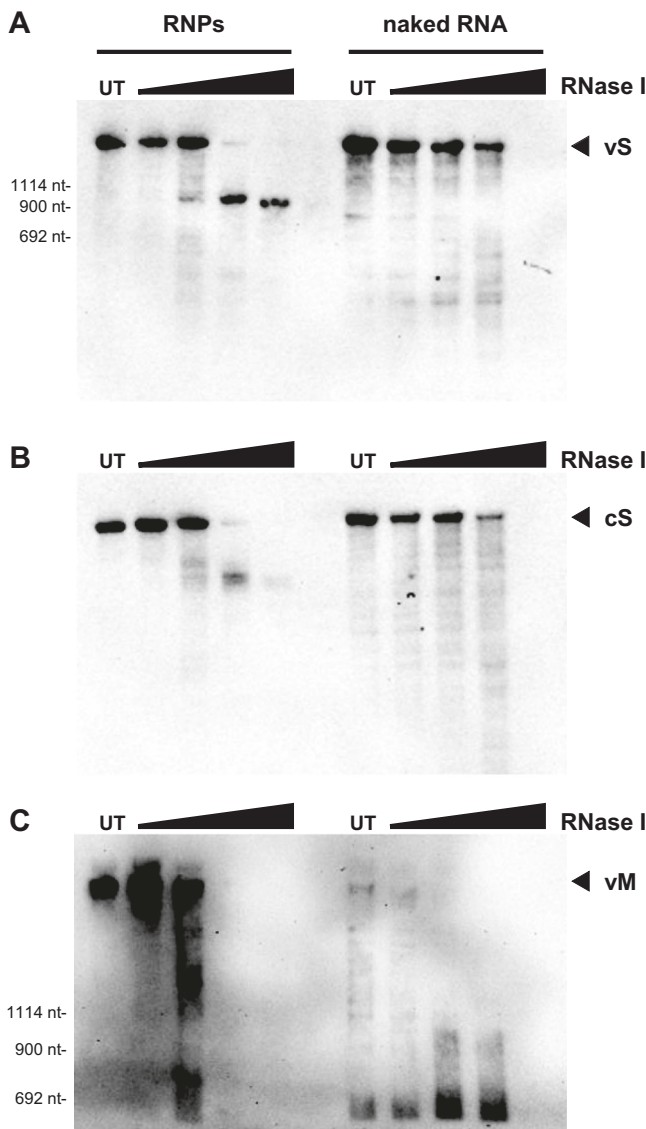

**Fig. 5 | Limited RNase I treatment reveals two-step mode of RVFV S segment RNP degradation.** Increasing amounts of RNase I (0.001, 0.01, 0.1, or 1 U/μl), compared to untreated control ("UT"), were applied for 3 min to either RNPs or to isolated total RNA from RVFV MP-12 virions produced in BHK-21 cells. Resulting RNA fragments were visualized by northern blot using the following DIG-labelled probes: **A** 3′ end S segment vRNA (vS) probe, **B** 3′ end S segment cRNA (cS) probe (see Fig. S5), and **C** M segment vRNA (vM) probe. Arrowheads indicate the intact genomic segments. Source data are provided as a Source Data file. The experiments in (**A**, **B**) were performed 3 times and those in (**C**) 2 times independently with similar results.

increasing RNase I concentrations (Fig. 5C). We conclude that RVFV S segment nucleocapsids contain a defined region in their middle that is sensitive to ssRNA-specific RNase digestion, indicating reduced or absent encapsidation.

**Confirmation of the exposed region in the genomic S segment**
To further validate the region of low N coverage on the genome S segment, we tested the accessibility by DNA antisense oligonucleotide (ASO)-targeted RNase H cleavage[23]. RNPs from RVFV particles were independently incubated with eight ASOs complementary to sequences of low or high N protein coverage (Fig. 6A, numbered black lines). After subsequent RNase H treatment, RNA was isolated and analyzed by northern blotting using the vS 3′end probe (see Fig. S5). Whenever an ASO is able to anneal to its target region, RNase H is expected to

cleave the RNA-DNA hybrid and therefore produce a northern blot band shorter than the input vS RNA. This was the case for ASOs 4, 5, and 6 covering sequences within the IGR, but also for ASO 3 that binds 5′ of ASO 4 outside of the IGR (Fig. 6B). By contrast, when using naked virion RNA as control, all eight ASOs directed cleavage by RNase H (Fig. 6C). Thus, the RNase H cleavage assay confirms and extends our results obtained by iCLIP, demonstrating the presence of a region with very low N coverage that we term the EvSR (for "exposed vS region"). Of note, although both iCLIP2 and RNase H protection assays of virion RNPs support an extended version of the EvSR, for infected cells the iCLIP2 analyses show comparatively high N coverage from the middle of the IGR on towards its 3′ end (Figs. 6A and S6A). Therefore, we conservatively define the region that is largely unencapsidated in both virions and infected cells (confined at the 3′ and 5′ ends by accumulations of N peaks) as the core EvSR (thick red line in Figs. 6A and S6A). Strikingly, in silico predictions indicated that both EvSR versions form an extended, imperfectly paired hairpin structure (Figs. 6D and S6B), whereas the canonical IGR between the two genes is predicted to form a large single-stranded loop closed by a terminal 8-bp stem (see Fig. S6B). The analyses also attested the predicted stem-loop structure a high likelihood (Fig. S7). On the antigenomic (cS) segment, the exposed regions detected by iCLIP2 (EcSR, "exposed cS region") were also overlapping with the IGR. In this case, both IGR and EcSR were predicted as being structured to a similar extent but in a different manner (Fig. S8). Taken together, our experiments have uncovered regions in the middle of the genomic and antigenomic S segments of RVFV that are largely void of N encapsidation and have the potential to form extended RNA secondary structures.

**Genome and antigenome S segments are not interacting in virus particles**
The presence of exposed regions in both the vS and cS nucleocapsids, along with the evidence that both S segments are co-packaged into RVFV particles, raised the question whether they interact via base pairing. To probe for such vS-cS interactions, we performed a Psoralen/UV RNA-RNA crosslinking assay coupled to northern blot analysis using DIG-labelled probes specific for the 5′ end regions of vS or cS RNAs (see Fig. S5, thick red lines). When naked virion RNA was used as control, the northern blot showed a major shift to RNA species of higher molecular weight, indicating the expected base pairing and crosslinking of vS and cS RNAs (Fig. 7 A, B). Nonetheless, neither isolated RNPs nor RNPs within virions exhibited such crosslinking. When RNA was crosslinked within infected cells, however, the shift was detected again. Thus, although the complementary genomic and antigenomic S segment nucleocapsids contain exposed regions in their center, they remain monomeric in the virus particles. The gel-shifting signal in infected cells likely indicates a transient interaction before packaging into virions or may result from vS/cS RNA interaction with their corresponding mRNAs. As a less likely alternative, the intracellular vS-cS crosslink signal could be due to the presence of non-encapsidated full-length RNAs in the cells. In any case, our data indicate that contrary to our expectations the vS and cS nucleocapsids do not hybridize inside RVFV particles.

**The EvSR is preserved in the NSs-deleted RVFV strain Clone 13**
We also applied the iCLIP2 method to Clone 13, a naturally attenuated RVFV mutant that had evolved in a nonfatal human case[24]. Clone 13 carries a large in-frame deletion that removes 69% of the coding sequence in the middle of the *NSs* gene, thereby inactivating its anti-host defense function[25,26]. Due to the deletion, the NSs of Clone 13 is a fusion of the 15 N-terminal and the 67 C-terminal amino acids, thus lacking 183 of the native 265 amino acids (Fig. 8A). Averaged iCLIP2 results from infected cells revealed N coverage patterns of the genomic and antigenomic segments that are similar to those of strain MP-12 (Fig. S9). Specifically, also Clone 13 exhibits non-encapsidated regions

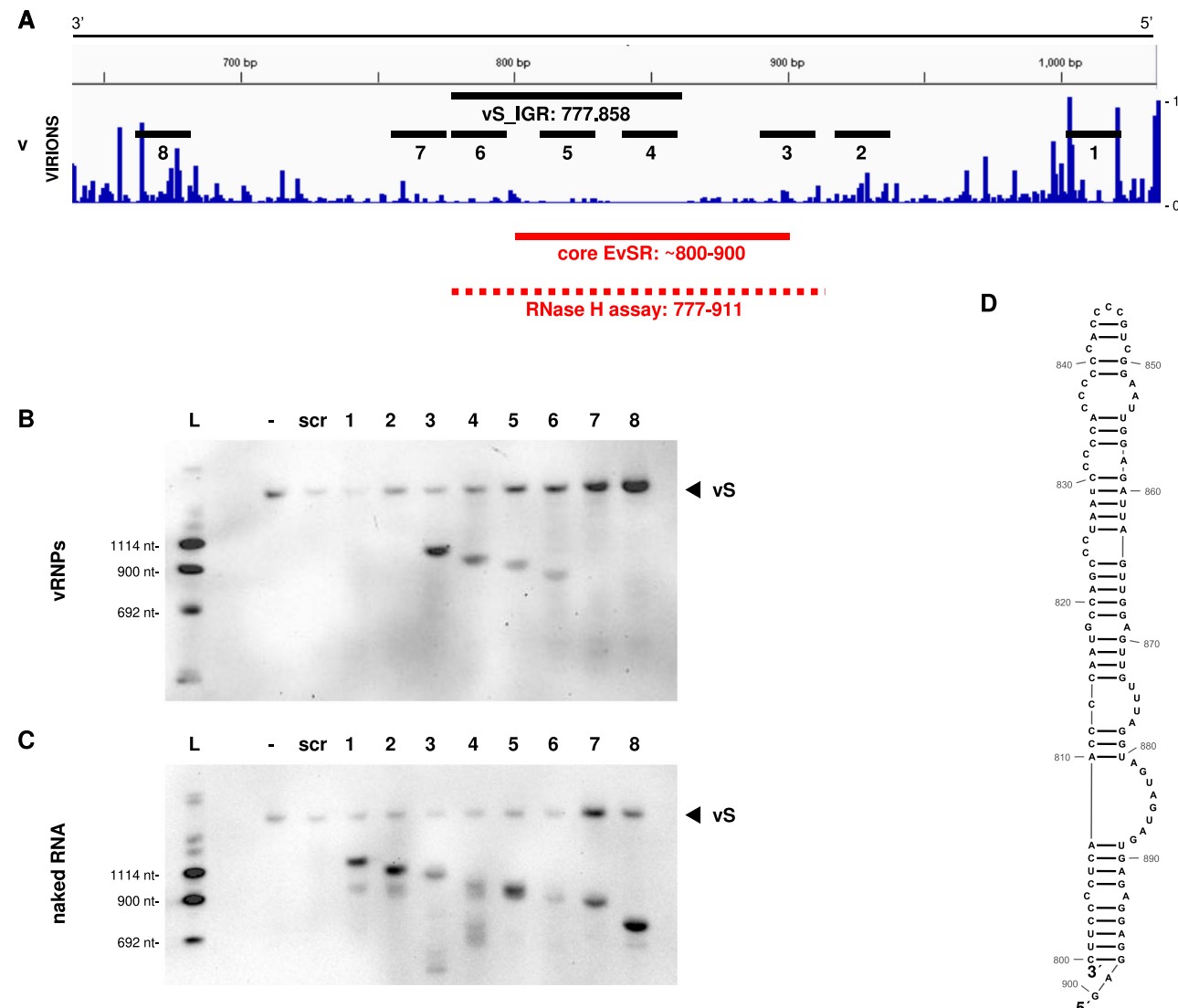

**Fig. 6 | Validation of genomic S segment RNA accessibility in RVFV MP-12 vRNPs.** Antisense oligonucleotide (ASO)-targeted RNase H digestion was performed using vRNP preparations or isolated RNA from RVFV MP-12 virions. **A** IGV tracks representing N binding profiles in the IGR-proximal region of the MP-12 vS segment (~650–1050), obtained for virions. iCLIP-derived crosslink sites for the genomic-sense RNA are shown here in 3′–5′ orientation and nucleotide positions are numbered as in the reference genome. Binding positions of targeting ASOs ("1–8") are illustrated, as well as the areas covered by the IGR (positions 777-858; black line) and the exposed vS region (EvSR, positions ~800-900; solid red line) derived from the iCLIP2 data. The region of the EvSR as inferred from the

accessibility experiment (from **B**) is shown as a dashed red line (positions 777-911). **B**, **C** RNase H digestion was performed in the presence of ASOs targeting the genomic S segment, or, as controls, without any ASO ("-" lanes) or in the presence of a scrambled oligonucleotide ("scr" lanes); vRNP preparations were used in (**B**), and isolated RNA from RVFV MP-12 virions in (**C**). Resulting RNA fragments were visualized by northern blot using the DIG-labelled probe 3′ end vS. The intact genomic segment ("vS") is indicated with an arrowhead. **D** RNA minimal free energy secondary structure by RNAfold for the proposed exposed vS region (core EvSR; corresponding to the solid red line in **A**). Source data are provided as a Source Data file. The experiments were performed 3 times independently with similar results.

in the vS and cS segments that start roughly in the middle of the IGR and extend into the terminus of the reading frame that locates downstream of the IGR (*NSs* for the vRNA and *N* for the cRNA) (Fig. 8B). For the antigenome, Clone 13 is similar to MP-12 because the EcSR extends into the *N* gene which is unchanged in Clone 13. The EvSR, however, is only maintained because NSs sequences that provide the downstream half of the EvSR are spared from the deletion in Clone 13. Indeed, structure predictions revealed high similarities to the EvSR of strain MP-12 (Fig. 8C; note that there are point mutations in the *NSs* 3′ region modulating the predicted structures).

The 3′ ends of the RVFV *N* and *NSs* mRNAs have been mapped previously, and the 3′-CGUCG-5′ motif identified as core transcription termination signal[8,27,28]. Interestingly, for both MP-12 and Clone 13 this core motif (red circles in Fig. 8C) is situated on top of the EvSR hairpin

structure whose 3′ half ("*N* side", on the left) is contributed by the IGR (transcribed as the noncoding 3′ tail of the *N* mRNA; circled blue nucleotides). The 5′ half of the hairpin ("*NSs* side", on the right) is provided by the counterstrand of the *NSs* reading frame (see Fig. 8C). Thus, both the sequence and the structure of the EvSR as well as the positioning of the *N* gene transcription stop signal are conserved in the naturally evolved *NSs* deletion mutant Clone 13, suggesting a role in the RVFV infection cycle.

**Functional testing of the EvSR**

We employed our reverse genetics systems[29–31] to directly investigate the function of the EvSR. Recombinant ambisense S segment reporter constructs were compared that contain the *N* gene in negative sense, and the Renilla luciferase (*Ren-Luc*) gene replacing the *NSs* gene on the

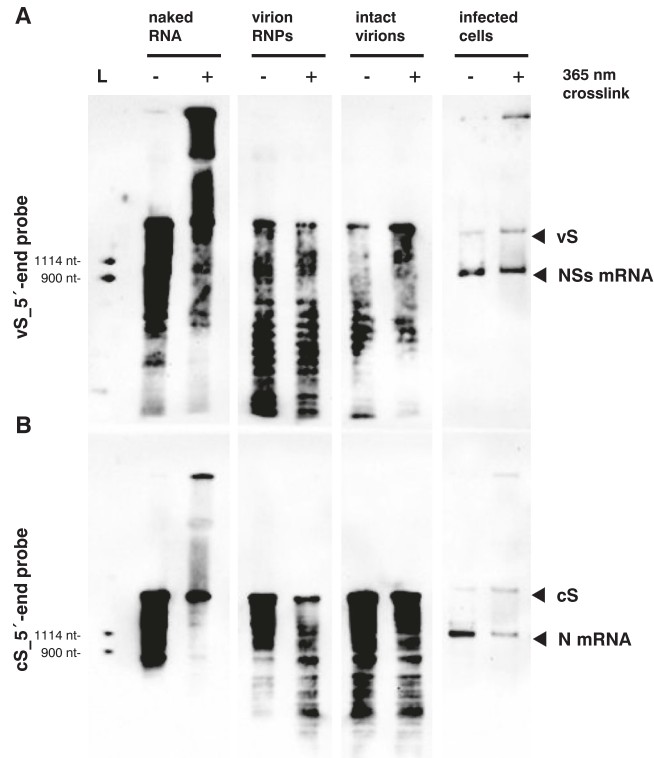

**Fig. 7 | RNA-RNA crosslinking and gel-shift assay to address the interaction between genomic and antigenic S segment RNAs.** For RVFV MP-12, isolated total RNA from virions, extracts containing vRNPs isolated from virions, intact virions or extracts containing vRNPs from infected BHK-21 cells (5 h p.i) were subjected to RNA-RNA crosslinking in the presence of Psoralen-azide at 365 nm. Resulting RNA fragments and band shifts were visualized by northern blot using the DIG-labelled probes **A** 5′ end vS or **B** 5′ end cS. Signals of the targeted genomic segments and mRNAs are indicated with arrowheads at the right margin. L, RNA ladder used as size marker. Source data are provided as a Source Data file. The experiments were performed 2 times independently with similar results.

opposing strand. The *N* and the *Ren-Luc* reading frames were thereby kept apart either by the IGR (which already contains part of the EcSR) or by the long version of the EvSR (Fig. 9A). To faithfully construct the latter, we added the EvSR-relevant part of the *NSs* sequence to the 3′ terminus of the *Ren-Luc* reading frame, separated by a stop codon to avoid potential interferences by *NSs* sequence translation. The respective S segment plasmid RVFV-vS-EvSR-Ren (containing EvSR 777-911) was then used to rescue a recombinant RVF reporter virus (see "Methods" section) which was compared with the IGR-only reporter virus we had constructed earlier[32]. The multistep growth curves in Fig. 9B show that the EvSR-containing virus (rRVFV-vS-EvSR-Ren) consistently produced approximately 100 times more progeny than the IGR-only virus (rRVFV-vS-IGR-Ren). Moreover, when we measured the infected cells, we observed that the EvSR-containing virus had about 10 times higher reporter activity at 24 h p.i. (Fig. 9C). These results indicate that the EvSR is a booster of RVFV multiplication. To narrow down the relevant step in the infection cycle, we employed transcriptionally competent virus-like particles (tc-VLPs). To produce the tc-VLPs, either of the two different S segment reporter plasmids was transfected into HEK293T cells together with helper plasmids encoding the RVFV polymerase L, the nucleoprotein N, and the glycoproteins (GPs). In the cells, the vRNA derived from the reporter plasmids is encapsidated by the co-expressed N protein and transcribed and replicated by the co-expressed L RdRP[30,31] (see also Fig. 1B, bottom). Figure 9D (columns 1 and 2) shows that the IGR S segment allows RVFV L-dependent (i.e. specific) reporter activity, but that

extension of the IGR to the wild-type EvSR led to superior activity (Fig. 9D, column 3, wt). As the tc-VLP reporter activity in the transfected cells is a measure of viral transcription and replication, we concluded that the EvSR enables a more efficient RNA synthesis of the S segment.

We also had mutated the basepairing nucleotides in the EvSR, either on the *N* side where the noncoding 3′ end sequence of the *N* mRNA is transcribed, on the *NSs* side where the *NSs* coding sequence is transcribed, or on both sides (see Figs. 9A and S10). As shown in Fig. 9D (columns 4–6) the mutants had an activity that was either close to the basal IGR reporter (*N* side mutant) or in between the IGR and the wt EvSR reporter activity (*NSs* side and *N/NSs* side mutants). Thus, the proper sequence (and most likely also structure) of the EvSR is important for viral transcription and replication.

As our set of expression plasmids had included the GPs, the transfected cells ("donor cells") presented in Fig. 9D are able to release tc-VLPs into the supernatants. The tc-VLPs can be harvested and their infection activity measured after transfer onto cells that express *L* and *N* ("indicator cells"). As shown in Fig. 9E, the properties of the individual S segment constructs are preserved after infection, but the difference between the IGR-only and the wt EvSR was even more pronounced (760 fold in donor cells vs. over 2000 fold in indicator cells), possibly suggesting that the EvSR also optimizes the packaging of RNPs into viral particles.

Our data with the various reverse genetics systems thus demonstrate that the exposed nucleotide sequence of the EvSR we had detected by iCLIP2 is important for optimal transcription and replication of the RVFV ambisense S segment.

In summary, we have mapped the interactions of the RVFV N protein with genomic and antigenomic RNA at single-nucleotide resolution. These analyses revealed that the vS and cS segments contain non-encapsidated elements in their central regions that overlap but are not identical with the IGR. The regions (EvSR and EcSR) have a high probability to form stem-loop structures and are accessible to complementary oligonucleotides. Moreover, the EvSR supported RVFV S segment transcription and replication substantially better than the IGR alone. The exposed regions EvSR and EcSR are therefore likely to exert a regulatory function in the RVFV replication cycle.

## Discussion

For negative-strand RNA viruses, nucleocapsids (i.e., RNA encapsidated by N and associated with the RdRP) rather than naked RNA serve as template for genome transcription and replication. In electron micrographs, the nucleocapsids of RVFV and other bunyaviruses appear as "beads on a string"[9,11–14]. However, we showed here by iCLIP2 analyses that encapsidation of genomic and antigenomic RVFV segments is not always regular but interrupted by regions of higher or lower N coverage. Specifically, iCLIP2 profiles obtained for the RVFV L and M segments suggest encapsidation over their entire segment lengths, as expected, whereas the genomic and antigenomic S segments contained a central region with very low to zero N coverage that was accessible to antisense oligonucleotide binding in RNase H protection assays. In silico structure predictions suggested that the exposed regions EvSR and EcSR with an approximate length of 100 nt form extended stem-loop structures. Moreover, reverse genetics experiments demonstrated that replacing the IGR with the EvSR provides superior replication and transcription activity of the S segment. Thus, the vS and cS nucleocapsids are interrupted in their central region by a functionally important stretch of naked and structured RNA.

The ambisense coding strategy, as utilized by phenuiviruses like RVFV, Severe fever with thrombocytopenia syndrome virus or tenuiviruses, arenaviruses like Lassa fever virus, and the plant-infecting tospoviruses[7], extends the genetic space by overcoming the "one segment - one gene" restriction that is normally imposed on

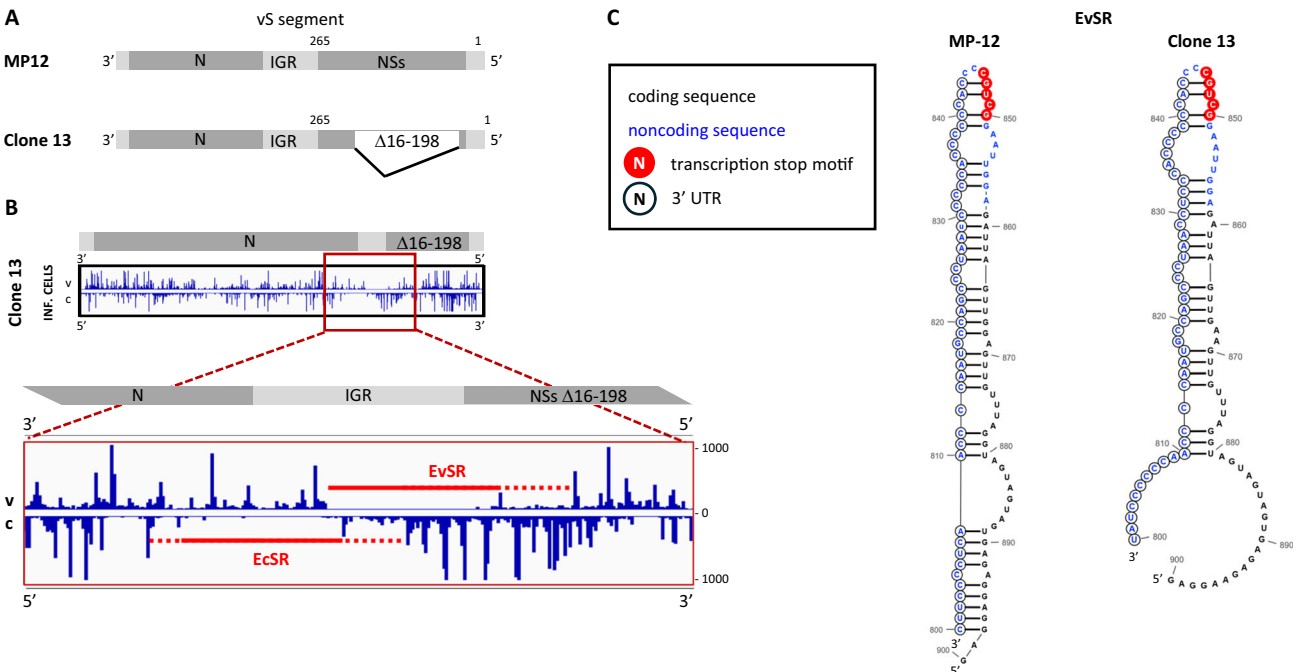

**Fig. 8 | The EvSR element is preserved in the NSs deletion mutant Clone 13.**
**A** Schematic representation of the genomic S segments of RVFV strains MP-12 (top) and Clone 13 (bottom). Numbers indicate the amino acid positions of NSs.
**B** Averaged iCLIP2 results of the Clone 13 genomic and antigenomic S segments (top) and magnification of the exposed regions in the center (bottom). Exposed

regions are indicated by a solid or dashed red line depending on the degree of N coverage. **C** Localization of the noncoding *N* mRNA sequences, coding *NSs* sequences, and the core transcription termination motif 5′-GCUGC-3′ (nucleotides in red cycles) within the EvSR stem-loop of MP-12 and Clone 13.

segmented negative-strand RNA viruses. Our iCLIP2 analyses suggest that the IGR, defined as the non-coding sequence between the ambi-sense ORFs, partially contributes to the non-encapsidated regions that we identified on the vS and the cS RNPs, but is also partially covered by N.

The previously identified core motifs 3′-CGUCG-5′ of the RVFV S segment transcription termination signals[8,27,28] are situated in the predicted hairpin structure, either on top (EvSR) or in the proximity of the top (EcSR) (Fig. 10A, B). Similar positionings of the core motif can be observed for the related phleboviruses Sandfly Fever Sicilian virus (SFSV) and Toscana virus (TOSV) when analogous regions (around 50 nt of IGR and 50 nt of coding sequence) were selected for in silico structure prediction (Fig. 10C, D). Perhaps, unwinding the stem-loop structures during transcription slows down the RdRP before it finally releases the template, as it is also indicated by the gradient of decreasing reads that were found towards the noncoding 3′ ends of the RVFV *N* and *NSs* mRNAs[8,27,28]. By any means, our data indicate that the position of the core termination motif on the exposed S segment RNA structures plays a role in RVFV mRNA transcription regulation.

These observations also raise questions regarding the replication of the full-length S segments. Firstly, it needs to be clarified how the exposed regions are generated. It is thought that during RNA replication, the nascent strand is directly encapsidated when leaving the active site of the RdRP[9,11,33]. In the RVFV nucleocapsids, each N monomer binds 4 nucleotides, and an additional 2–3 nucleotides are at the N-N interface[14]. With an EvSR/EcSR length of at least 100 nt, this would mean that either encapsidation by 14 or more N subunits is bypassed during replication, or that the N subunits are post-replicationally removed by an unknown process (which could even be caused by the formation of the dsRNA structures themselves, as N does not bind dsRNA[14]). Secondly, it remains to be investigated how the EvSR/EcSR can be transcribed or replicated since nucleocapsids (and not just RNA) are the template for the RdRP of negative-strand RNA viruses. It is however conceivable that short stretches of naked RNA are tolerated

as template. Thirdly, it is unclear how an exposed region in the middle of a nucleocapsid is regulated to either terminate mRNA transcription or bypass the mRNA termination signal for genome replication. Conventionally such a switch between transcription and replication is thought to be triggered by the absence or presence of N protein, respectively[34], which in the case of the non-encapsidated EvSR and EcSR does not apply. Apparently, the decision to terminate transcription or continue replication is subject to another control mechanism independent of N availability, e.g., flexible folding/unfolding of the stem loop. Fourthly, the exposed RNA structures are expected to render the virus vulnerable to attacks by cellular RNases and innate immunity sensors. Indeed, the antiviral RNA helicase DDX17 has been shown to bind to RVFV M and S nucleocapsids, the latter in a region that overlaps with the IGR[35]. Moreover, regions within the EvSR and EcSR of RVFV are vulnerable to small interfering RNAs[36,37], and similar observations were reported for the plant-infecting Tospo bunyavirus[38]. The multiple and strong anti-innate immunity functions exhibited by the RVFV protein NSs[39,40], however, are expected to at least partially dampen these antiviral attacks.

For the segmented negative-strand RNA virus FLUAV it was shown that the incorporation of the eight nucleocapsids into progeny particles is selective and depends on base pairing between packaging signals on individual segments[41,42]. In line with this, nucleotide resolution mapping of the FLUAV nucleoprotein (NP)-RNA interactions revealed vRNA regions of poor NP coverage on most of the genome segments[15,16]. As mentioned, FLUAV is however unusual since in general the RNA bases are not hidden in the NP but exposed to the solvent[6,19]. Although our iCLIP2 analyses of RVFV did not detect large exposed regions on the L or M segments and most of the S segments, we cannot exclude the existence of minor protruding loops of RNA mediating segment-segment interactions.

The WHO had designated RVF as one of its eight priority diseases[43]. However, despite the recognition as a significant threat to public health and the economy in the affected regions, there are no

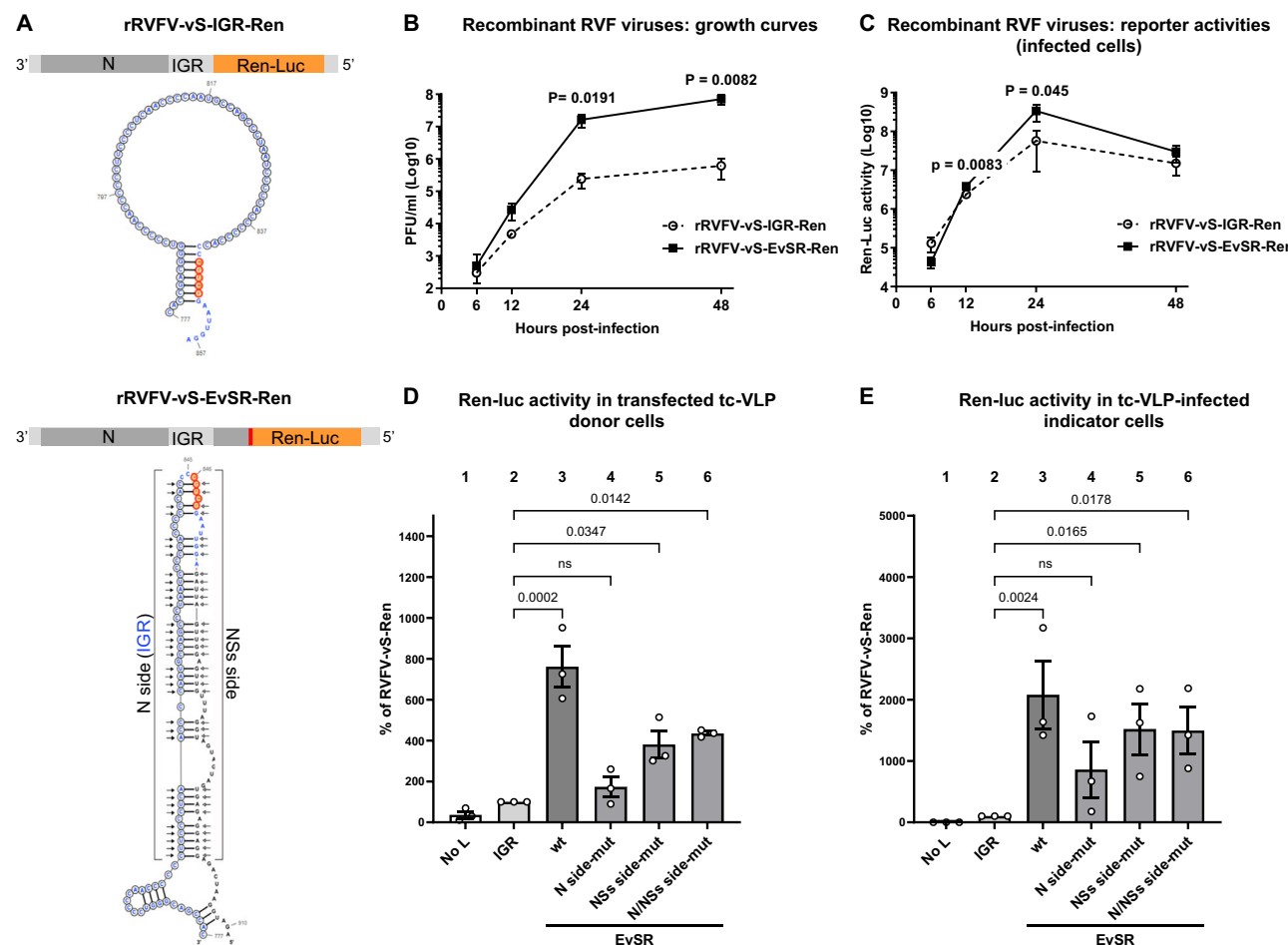

**Fig. 9 | Functional relevance of the EvSR. A** Schematic representation of the genomic S segments of the recombinant RVFV strains rRVFV-vS-IGR-Ren (top) and rRVFV-vS-EvSR-Ren (bottom) with their predicted RNA structures underneath. The nucleotides that were changed in the various mutants are indicated by arrows (for details see Fig. S10). In the reporter S segment cartoons, the orange bar indicates the Renilla luciferase (Ren-Luc) gene and the red vertical line cartoon indicates the stop codon of the *Ren-Luc* gene. Symbols in the predicted structures are as in Fig. 8. **B**, **C** Multistep growth curves. BHK-21 cells were infected with the recombinant RVF viruses at an MOI of 0.01. At the indicated time points, infectious viruses in supernatants were titrated (**B**), and Ren-Luc activities in the cells measured (**C**). **D** tc-VLP donor cells. HEK293T cells were transfected with expression plasmids for RVFV L, N, and the GPs and either of the reporter S segments (including the EvSR mutants) depicted in (**A**). Ren-Luc reporter activity was measured at 72 h post transfection. **E** tc-VLP indicator cells. HEK293T cells, pretransfected with the RVFV L and N expression plasmid, were incubated with the indicated tc-VLP-containing supernatants, and Ren-Luc activity was measured 24 h later. In **D** and **E** all Ren-Luc activities are shown relative to the tc-VLPs with the IGR reporter segment which were set to 100%. In (**B–E**), mean values and SEM of 3 independent biological replicates with 3 technical replicates each are shown. Two-tailed unpaired *t*-tests (**B**, **C**) and one-way ANOVA with Dunnett's correction for multiple comparisons (**D**, **E**) were used for statistical testing: *P* values above the significance threshold of 0.05 are indicated in the graphs.

licensed live-attenuated vaccines for human use[44]. One concept to increase the safety and immunogenicity of RVFV vaccines is to either delete the gene for the virulence factor NSs[45,46], or to replace it with the *NSs* of a related phlebovirus of lower pathogenicity[47,48]. Moreover, *NSs*-deleted RVFV recombinants expressing a reporter gene or a foreign *NSs* are convenient tools to measure interferons in a species-independent manner[32], screen for antivirals[49], or investigate the function of NSs proteins for which no reverse genetics systems are available[47,48,50,51]. Our results with tc-VLPs and recombinant RVFV suggest that all these strategies using the *NSs* locus as insertion site for genes of interest can be improved by introducing the EvSR into a recombinant RVFV.

In conclusion, we mapped the entire N-vRNA and N-cRNA interactions of RVFV and established that the vS and cS segments possess at least 100 nt long non-encapsidated regions (EvSR and EcSR) in their center with about 50 nt covering the downstream half of the IGR and 50 nt covering the terminal sequence of the adjacent downstream gene. The exposed regions have a high propensity to form defined stem-loop structures and harbor the transcription termination signals of the genes that are expressed on the respective segment. Moreover, the EvSR was found to substantially improve S segment activity as compared to the canonical IGR. Due to the apparent regulatory role in the viral replication cycle, it is expected that phleboviruses in general, as well as other ambisense viruses like bandaviruses, tenuiviruses and arenaviruses, possess similar non-encapsidated regions in their nucleocapsids.

## Methods
### Cells and viruses
Human Lung Carcinoma cell line A549 (ATCC) and Baby Hamster Kidney (BHK-21) cells (ATCC) were maintained in Dulbecco's modified Eagle's medium (DMEM) supplemented with 10% fetal bovine serum (FBS) and Penicillin/Streptomycin (all from Thermo Fisher Scientific) in a 5% $CO_2$ atmosphere at 37 °C.

Rift Valley Fever Virus (RVFV) strain MP-12 (GenBank entries DQ380154.1; DQ380208.1; DQ375404.1) and Clone 13 (GenBank entries DQ380182.1; DQ380213.1; DQ375417.1) were grown under BSL-2 conditions on BHK-21 cells for 48–72 h following inoculation with virus at MOI = 0.0005 for 1 h at 37 °C.

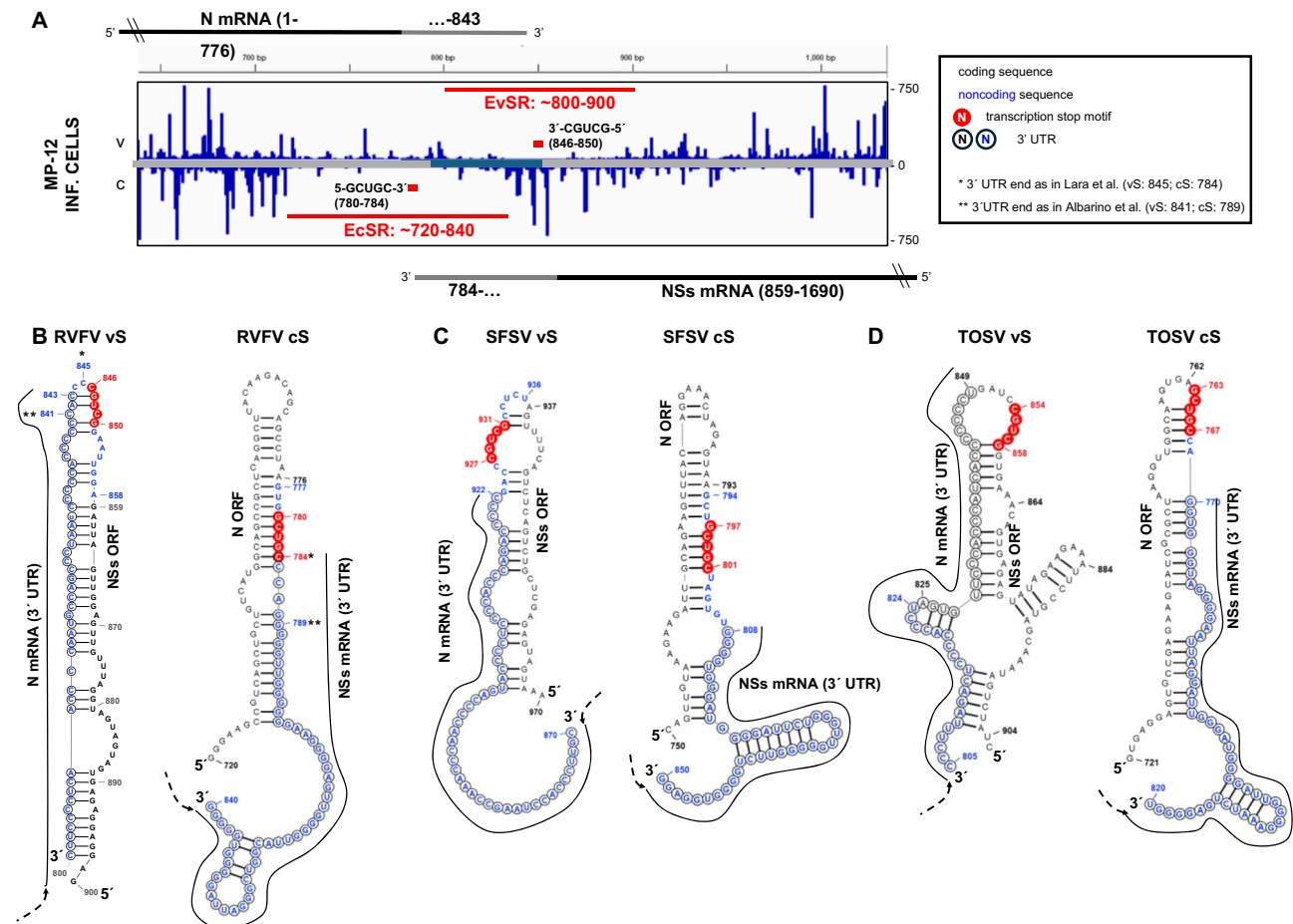

**Fig. 10 | EvSR and EcSR structures and the regulation of RVFV mRNA transcription termination. A** Averaged N protein iCLIP2 profiles for the IGR and surrounding exposed regions along vS and cS segment RNAs. The location of the core transcription termination motif 5′-GCUGC-3′ is marked by short red boxes at vS and cS strands. Coding (gray) and untranslated (blue) regions are indicated in the center of the profile. Corresponding mRNAs are represented as black lines with their 3′ untranslated regions (UTRs) shown as gray lines at the top and bottom of the panel. **B–D** Localization of the core transcription termination motif 5′-GCUGC-3′ (nucleotides in red cycles) within or close to the apical region of the predicted hairpin structures of RVFV EvSR and EcSR (**B**) and the corresponding regions of Sandfly Fever Sicilian virus (SFSV) (**C**) and Toscana virus (TOSV) (**D**). Nucleotides of the coding regions are depicted in black and those of the IGR in blue. 3′ untranslated regions for corresponding mRNAs are shown as nucleotides in cycles and the mRNAs are depicted with solid black lines. The 3′UTR mapping for RVFV matches the findings of Ikegami et al.[27], and * indicates the mapping data of Lara et al.[28] and ** those of Albarino et al.[8]. The mRNA 3′UTRs of SFSV and TOSV were mapped by Albarino et al.[8]. The structures for the genomic S segments are shown from 3′ to 5′ and those for the antigenomic S segments from 5′ to 3′.

## Virus infection and sample preparation

For N-CLIP from virions, preparation, and collection of RVFV MP-12 virus particles were performed as follows. BHK-21 cells were infected and incubated as described above and the inoculum was replaced by DMEM with 5% FBS to improve virus concentration when using filter units. Virus-containing supernatant was then collected and cleared from cell debris by centrifugation (Beckmann, rotor SX4750) at 2000 × g for 10 min. Next, the virus suspension was concentrated 20–40 fold using Amicon Ultra-15 (Millipore; article # UFC901008D) concentration units at 4000 × g for 15 min; at this step 25% input sample (equivalent to one iCLIP2 sample) was collected prior to the crosslinking procedure. The concentrated suspension was transferred into a new culture dish on ice and subjected to UV-C irradiation (twice with 300 mJ/cm² at 254 nm) using a BLX-254 UV-Crosslinker (Bio-Link). After crosslinking, the virus suspension was diluted in 1 volume of PBS and once again concentrated 10-fold using Amicon Ultra-4 (Millipore; article # UFC8010D) concentration units. Thereby, each iCLIP2 sample was derived from at least a 100-fold concentrated virus suspension. The resulting suspension was lysed with 9 volumes of PXL buffer (1× PBS, 1% IGEPAL CA-630 (Sigma–Aldrich, product # I8896), 0.5% deoxycholate, 0.1% SDS) for 10 min at 4 °C. Commonly, one

iCLIP2 sample eventually contained $3 \times 10^8$ plaque-forming units (PFU) of RVFV MP-12, as determined by plaque assay (see below) from non-crosslinked control samples.

For N-CLIP with infected BHK-21 cells (with RVFV MP-12) or A549 cells (with RVFV Clone 13), the virus was inoculated at MOI = 1. Then, 5 h p.i cells were washed twice with cold PBS, UV-C-irradiated (400 mJ/cm² at 254 nm) and lysed in 1 ml RIPA buffer (50 mM Tris-Cl pH 7.4), 1% IGEPAL CA-630, 0.1% SDS, 0.25% desoxycholate, 150 mM NaCl, 5 mM EDTA, 1× Protease Inhibitor cocktail (c0mplete, Roche, Cat # 4693116001) per 15-cm culture dish. Here, one iCLIP2 sample corresponded to $1–2 \times 10^7$ of infected cells. All N-iCLIP2 experiments from with virions and infected cells were performed in 3 independent biological replicates.

All infection experiments were conducted under biosafety level 2 conditions. Quantification of infectious virus was achieved by plaque assay on BHK-21 cells (see below).

## RVFV nucleoprotein iCLIP2 and RNA-Seq library generation

We developed a protocol for mapping N-RNA interactions of RVFV[52], utilizing a second-generation version of the iCLIP method[21], termed "iCLIP2"[20]. For both N-iCLIP2 with virions and MP-12-infected BHK-21

cells, three experimental samples were designed: for immunoprecipitation with two different mouse monoclonal antibodies (mAbs) against RVFV N, (i) A9F12 (antibody 1[53]) and (ii) 10A7 (antibody 2; G. Lorenzo, unpublished), and for immunoprecipitation (iii) with a control antibody (anti-beta-actin mouse monoclonal ab8226, Abcam, or an anti-HA.11 epitope tag mouse monoclonal antibody cat # 901515, Biolegend, respectively). For N-CLIP from Clone 13-infected A549 cells, RVFV N-specific mouse mAb 2B1 (antibody 3) (G. Lorenzo, unpublished) was utilized.

Following preparation from RVFV MP-12 virions or infected BHK-21 or A549 cells (see above), extracts were subjected to RNase treatment using RNase I (Ambion, cat # AM2294) at an empirically determined concentration in the presence of 4 U/ml of Turbo DNase (Ambion, cat # AM2238) and 40 U/ml of RNaseOUT (Thermo Fisher Scientific, cat # 10777019). For N-iCLIP2 with virions, the best results were obtained when two RNase I concentrations (0.1 U/μl and 0.01 U/μl) were applied in parallel treatments, whereas for N-iCLIP2 with infected cells, 0.01 U/μl RNase I was applied to the entire lysate volume. After incubation for 3 min at 37 °C, RNA digestion was terminated on ice and extracts were immediately subjected to immunoprecipitation for at least 2 h at 4 °C. After washing four times with washing buffer (50 mM Tris-Cl pH 7.4, 1 M NaCl, 0.05% Tween 20), the co-immunoprecipitated RNA was dephosphorylated, ligated to a 3' RNA linker and 5'-radiolabeled with T4 Polynucleotide Kinase (NEB, cat # M0201S) and [γ-$^{32}$P]-ATP. Samples were subjected to neutral pH SDS-PAGE (NuPAGE, Thermo Fisher Scientific, cat # NP0321) and transferred to a nitrocellulose membrane (VWR, cat #10600004). RVFV N/RNA-complexes were visualized by autoradiography and then excised from the membrane (at this step low and high RNase I samples for virions were combined). Subsequently, the protein was digested with 1 μg/μl Proteinase K (Carl Roth, oder # 7528.1) and the eluted RNA was subjected to iCLIP2 library preparation as previously described[20].

Quality and quantity of amplified cDNA were assessed with the High Sensitivity DNA Kit (Agilent, order # 5067-4626). Libraries were pooled, denatured, and diluted to 4 nM. For libraries from RVFV virions, paired-end reads were generated with the MiSeq Reagent Kit v2 Micro (300 cycles) (Illumina, cat # MS-103-1002) reagents and flow cell. Libraries from RVFV-infected BHK-21 cells were sequenced with the MiSeq Reagent Kit v3 (150 cycles) (Illumina, cat # MS-102-3001), generating paired-end reads. For iCLIP2 libraries from RVFV-infected A549 cells, single-end reads were sequenced with the NextSeq 500/550 Mid-Output Kit (150 cycles) (Illumina, cat # 20024904) using a loading concentration of 1.8 pM. The input libraries were generated either from non-crosslinked material (for RVFV virions, using the NEBNext Kit for Illumina) (NEB, cat # E7370L) and MiSeq Reagent Kit v2 Micro (300 cycles) or as so-called size-matched inputs – for infected BHK-21 (together with iCLIP2 libraries) and A549 cells (separately from iCLIP2 libraries, using the MiSeq Reagent Kit v2 (300 cycles) in paired-end mode), as initially described in the eCLIP protocol[54]. The latter approach includes all RNAs that were crosslinked to any protein in the same size range as RVFV N/RNA-complexes.

In all executed sequencing processes, 1% spike-in of the PhiX Control library (Illuina, cat # FC-110-3001) was added according to the manufacturer's guidelines. For each run on the MiSeq system, the loading concentration was 10 pM. After image processing, base calling, and demultiplexing of sequenced reads, fastQ-files were obtained. Sequencing adapter barcodes are listed in Supplementary Tables S1–S3.

### Next-generation sequencing data analysis
The RVFV reference genome was retrieved from NCBI[55] for strains MP-12 (accession numbers DQ380154.1, DQ380208.1, and DQ375404.1) and Clone 13 (accession numbers DQ380182.1, DQ380213.1, DQ375417.1).

iCLIP2 data analysis was performed with PARANOiD[56] applying the default settings. Briefly, PARANOiD performed quality control and trimming based on FastQC (version 0.11.9; https://www.

bioinformatics.babraham.ac.uk/projects/fastqc/), TrimGalore (version 0.6.7; https://www.bioinformatics.babraham.ac.uk/projects/trim_galore/) and the FASTX toolkit (version 0.0.14; https://github.com/agordon/fastx_toolkit), barcode extraction and deduplication was carried out with umi_tools (version 1.1.4)[57], and reference mapping - with bowtie2 (version 2.5.1)[58]. Low-quality and multimapped reads were filtered out and crosslink sites were assigned to 1 nucleotide upstream of the start position for forward-orientated alignments and to 1 nucleotide downstream of the end position for reverse-oriented alignments.

RNAseq reads were first adapter trimmed via TrimGalore and then filtered from low-quality bases via fastq_quality_filter from the FASTX toolkit, removing bases with a quality score below 20 and reads with less than 90% of their bases above that quality score. Remaining reads were aligned via Bowtie2 to the MP-12 reference genome and all alignments with a MAPQ-score below 2 were removed. Resulting alignments were summarized for their assigned fragment and their strand orientation and the resulting counts normalized using RPKM as follows: $\frac{\text{number of reads for current strand } \textit{and}/\textit{or} \text{ fragment}}{(\text{fragment length}/1.000)*(\text{total number of reads}/1.000.000)}$.

Fragment lengths were used according to the reference of MP-12 which are: 1.690 nt for DQ380154.1, 3.885 nt for DQ380208.1, and 6.404 nt for DQ375404.1.

To summarize replicates PARANOiD calculated the arithmetic mean for every position. To provide a statistical measurement of the similarity of the data a correlation analysis was performed using R (version 4.0.3)[59]. Results were visualized using IGV (Integrative Genomics Viewer)[60].

### Northern blot analysis
For the purpose of strand-specific visualization of RVFV vS and cS RNAs by northern blot, DIG-labeled RNA probes complementary to each terminus of the RVFV vS and cS RNA were designed (designated as 5' end and 3' end probes, see Fig. S5). PCR templates for these probes were generated by Phusion DNA Polymerase (NEB, cat # M0530S) using plasmid pHH21_RVFV_vS[29] encoding for RVFV ZH-548 S segment sequence (DQ380151.1) and primers listed in Table S4. The resulting in vitro transcription templates for SP6 RNA polymerase (NEB, cat # M0207S) were gel-purified (Omega, cat # D2500-01) and utilized for synthesis of internally DIG-labeled probes using the Roche DIG-RNA labeling kit (cat # 11745832910). Probes were denatured at 95 °C for 2 min before use. As a control, a DIG-labeled RNA probe for the RVFV vM segment was designed and generated using the same strategy.

RNA samples were denatured in formamide loading buffer at 95 °C for 2 min and separated by denaturing 6% PAGE (2.5 h at 200 V for S and 4 h at 200 V for M segment detection) together with the DIG-DNA ladder VIII (Roche, cat # 11669940910). Transfer to Nylon transfer membrane (GE Healthcare, cat # RPN1210B) was conducted via semi-dry blotting for 1 h at 3 mA/cm². Then, the membrane was crosslinked at 120 mJ/cm² (254 nm) and prehybridized for 1 h in HybBuffer (Roche, cat # 11603558001) at 65 °C. Hybridization was performed in a hybridization oven at 65 °C for at least 24 h with a probe concentration of at least 100 ng/ml. Following hybridization, washing, and detection of DIG-labeled RNA targets were performed using the Roche DIG Northern Starter Kit (cat # 12039672910) according to the manufacturer's instructions. Chemiluminescence was detected with a ChemiDoc imaging system (BioRad).

### Limited RNase digestion in vitro
To observe a distinct pattern of vRNP degradation in comparison to naked RNA, the samples were subjected to digestion with increasing concentrations of RNase I (Ambion). Infections were performed according to the iCLIP2 conditions, as described above. Collected virus-containing supernatant was concentrated

around 100-fold prior to lysis with 4 volumes of lysis buffer (60 mM HEPES, pH 8.0, 300 mM KCl, 15 mM MgCl$_2$, 5% IGEPAL CA-630, 25 U/ml of RNaseOUT) for 30 min on ice. For total RNA isolation, cells were lysed with TRIzol (Thermo Fisher Scientific, cat # 15596026) following the manufacturer's instructions. Naked RNA isolated from concentrated virions was resuspended in nuclease-free H2O.

For RNase digestion, both vRNP extract and purified RNA were supplemented with 9 volumes of RQ1 buffer (40 mM Tris-Cl pH 8.0, 10 mM MgSO$_4$, 1 mM CaCl$_2$) and RNase I at concentrations 0, 0.001 U/μl, 0.01 U/μl, 0.1 U/μl or 1 U/μl (in presence of 4 U/ml of Turbo DNase and 40 U/ml of RNaseOUT). Following incubation for 3 min at 37 °C and agitation at 800 rpm, samples were transferred on ice and RNA was isolated by phenol/chloroform extraction (Carl Roth, cat # A159.1) and ethanol precipitation. Lastly, RNA was dissolved in formamide loading buffer and subjected to northern blot analysis (as described above) and detection using the DIG-labeled probes 3' end vS, 3' end cS, and vM.

## RNA-RNA crosslinking
To address the potential interaction between RVFV segments, RNA-RNA crosslinking was performed. BHK-21 cells were infected with RVFV MP-12 either at MOI = 1 for 5 h, or at MOI = 0.0005 for 48 h. Supernatant collected from the latter was subjected to virion concentration (as described above) and then divided into three parts. One part was utilized for RNA isolation using TRIzol, another – for vRNP isolation using 1× PBS supplemented with 0.5% Triton X-100, and the third part remained untreated at this step. The volume for all samples was adjusted with PBS, each sample was then divided into two halves, supplementing one-half sample with 50 μg/ml of Psoralen-TEG azide (Berry&Associates, cat # 1352815-11-2), followed by crosslinking with 5 J/cm$^2$ at 365 nm on ice; the second half sample was kept as a non-crosslinked control. For each sample, the untreated and crosslinked half samples were subjected to Proteinase K treatment (1 mg/ml) for 15 min at 42 °C, followed by phenol/chloroform extraction and ethanol precipitation.

The RNA-RNA crosslinking in infected cells was performed as follows. Cells were washed twice with PBS, followed by the addition of Psoralen diluted to 500 μg/ml in OptiMEM (or OptiMEM without psoralen for control cells) and incubation for 20 min at 37 °C. Then, cells were subjected to crosslinking with 5 J/cm$^2$ at 365 nm on ice, and the psoralen solution was subsequently aspirated. Cells were immediately lysed in RLT buffer (Qiagen, cat # 79216) supplemented with 40 mM DTT and total RNA was isolated according to the manufacturer's instructions. Resulting RNA samples for naked RNA, vRNPs, intact virions, and infected cells (crosslinked or not) were analyzed by northern blot using the DIG-labeled probes 5' end vS and 5' end cS for detection.

## ASO-targeted RNase H digestion in vitro
Validation of RVFV vRNP accessibility based on iCLIP2 data was addressed by ASO-directed RNase H digestion experiments in vitro[23] using preparations of RVFV MP-12 virions. Infections were performed according to the iCLIP2 conditions (see above), and viral RNP isolation from virions – as outlined for RNase I digestion (see above). At least 0.5–1 × 10$^8$ PFU of RVFV MP-12 corresponded to one RNase H digestion reaction. ASO-mediated RNase H cleavage reactions were performed at 37 °C for 30 min in parallel for vRNP and vRNA (0.1 μg RNA per reaction) samples using 5 μM ASO and 2 U RNase H (NEB, cat # M0297S) in a total volume of 50 μl. As control reactions, RNase H digestion in the presence of a scrambled DNA oligonucleotide and a reaction in the absence of any ASO and RNase H were conducted. The ASOs and control oligonucleotide used are listed in Table S5. Finally, reactions were terminated by TRIzol addition and isolation of RNA. Resulting RNA samples were redissolved in formamide

loading buffer and visualization of cleaved viral RNA fragments was achieved by northern blotting using 3' end vS probe.

## Reverse genetics systems of RVFV
The IGR S segment reporter plasmid pHH21-RVFV-vS-[IGR-MP-12]-Ren contains the MP-12 point mutation C829U in the IGR. This was also used as template to generate S segment plasmid pHH21-RVFV-vS-EvSR-Ren, containing the *Ren-Luc* ORF fused 3' terminally to the 3' terminal 54 nt of the *NSs* gene to maintain the integrity of the EvSR. A stop codon was inserted at the end of the *Ren-Luc* ORF to prevent the production of a fusion protein. Using oligonucleotides and an In-Fusion recombination kit (Takara Bio Inc, cat # 102518), mutations of the EvSR (see Supplementary Fig. S10) were set either on the *N* side (nt 800–844) or the *NSs* side (nt 848–899) by exchanging base-pairing nucleotides with the respective other purine or pyrimidine.

VLP assays were carried out as previously described[30,31] with slight modifications. Briefly, HEK293T cells were seeded at 3 × 10E5 cells/well in 6 well plates and transfected the following day with support plasmids encoding RVFV N, M, and L (pI.18-RVFV-N, pI.18-RVFV-M, pI.18-RVFV-L) as well as dominant negative PKR (pI.18-HA-PKRdelE7)[31] and the firefly luciferase (*FF-luc*) expressing transfection control (pGL3-Luc). The above-described S segment EvSR reporter (pHH21-RVFV-vS-EvSR-Ren), its *N*-side and *NSs*-side mutants, or the IGR-only S segment reporter pHH21-RVFV-vS-Ren[32] were used as minigenome constructs. After 72 h, supernatants were harvested and the cells lysed in passive lysis buffer (Promega, cat # E1941). The harvested supernatants were applied to HEK293T cells previously transfected with plasmids expressing RVFV *N* and *L* as well as dominant negative PKR. Lysates were harvested in passive lysis buffer 24 h post-supernatant treatment. Using the dual-luciferase reporter assay system (Promega, cat # E1910) and a Centro LB 960 microplate luminometer (Berthold Technologies, cat # 38100-50) the Ren-Luc and FF-luc activities were measured in both sets of lysates.

The previously described polI/II RVFV rescue system[29] was used to generate a recombinant RVFV virus with an EvSR-containing S segment expressing *Ren-Luc*. To this end, 1:1 co-cultures of HEK293T and BHK-21 cells were transfected with plasmids for RVFV N and L (pI.18-RVFV-N and pI.18-RVFV-L), the full M and L genome segments (pHH21-RVFV-vM and pHH21-RVFV-vL), as well as the above-described pHH21-RVFV-vS-EvSR-Ren for the S segment. Supernatants were harvested at 5 days post-transfection and used to grow virus stocks on Vero E6 cells. The resulting virus was designated RVFV-vS-EvSR-Ren.

## Determination of virus titers
Supernatants with MP-12 or Clone 13 were titrated by a conventional plaque assay on confluent monolayers of BHK-21 cells in 12-well culture plates. Serial dilutions of culture supernatants were prepared in serum-free Opti-MEM medium on ice, followed by inoculation of BHK-21 cells with 150 μl of each dilution at 37 °C. After 1 h, the inoculum was replaced with Cellulose overlay (3% Cellulose, 5% FBS, Penicillin/Streptomycin in 1× MEM) and cells were incubated for 72 h at 37 °C. Following the overlay removal and washing with PBS, monolayers were fixed and stained with Crystal Violet solution (0.75% Crystal Violet, 3.75% formaldehyde, 20% ethanol, 1% methanol) for 10 min at room temperature. Virus titers in corresponding samples were determined through quantification of PFU in relevant dilutions in at least 3 biological replicates.

The *Ren-Luc*-expressing reporter viruses were titrated by immunoplaque assay[61]. Briefly, the infected cells were incubated for 48 h, fixed, permeabilised (PBS/0.5% Triton X-100), blocked (PBS/5%BSA), and stained using mouse anti-RVFV-N monoclonal antibody mouse 1B2 (Benjamin Lamp and F. Weber, unpublished) followed by goat anti-mouse IgG IRDye® 680RD secondary antibody (LiCor Bioscience, cat # 926-68070). Plaques were visualized on a LiCor Odyssey M (cat # 3350-STA).

## RNA structure prediction

RNA shape class probabilities were calculated using the "RNAshapes" tool in probability mode[62] via the BibiServ Bioinformatics Service (https://bibiserv.cebitec.uni-bielefeld.de/). Representative structures, those with the best free energy within a class, for top probable shape classes were visualized using the online version of "forna" from the ViennaRNA package (http://rna.tbi.univie.ac.at/cgi-bin/RNAWebSuite/RNAfold.cgi) or RNAcanvas web service (https://rnacanvas.app/)[63] applying structure constraints provided by "RNAshapes". Minimum free energy (MFE) structure drawings were generated using "forna" package, which provides a web interface for displaying RNA secondary structures using the force-directed graph layout provided by the d3.js visualization library. Ensemble base pair probabilities were visualized as Dot Plots (upper right triangle), whereas the lower left triangle indicates base pair probabilities that can compatible integrated into a single secondary structure (maximum accuracy structure), i.e. do not cross.

## Statistical analysis

To compare the effects of IGR or EvSR manipulations in reverse genetics systems, statistical testing was performed as indicated, using GraphPad Prism version 9.3.1 and 10.2.2 for Windows (GraphPad Software, San Diego, California USA, www.graphpad.com).

## Reporting summary

Further information on research design is available in the Nature Portfolio Reporting Summary linked to this article.

## Data availability

The iCLIP2 read data generated in this study have been deposited in the GEO database under accession numbers ERP153262 (complete study), ERR12252313 (RNAseq rep1), ERR12252314 (RNAseq rep2), ERR12252315 (RNAseq rep3), ERR12252873 (iCLIP virions), ERR12252875 (iCLIP MP-12/BHK), ERR12253852 (iCLIP clone13/BHK). Viral sequences were downloaded from Genbank entries DQ380154.1 (S segment RVFV MP-12), DQ380208.1 (M segment RVFV MP-12); DQ375404.1 (L segment RVFV MP-12), DQ380182.1; (S segment RVFV Clone 13), DQ380213.1 (M segment RVFV Clone 13), DQ375417.1 (L segment RVFV Clone 13). Source data are provided with this paper.

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

## Acknowledgements

This work is supported by the Deutsche Forschungsgemeinschaft (DFG) grants SFB 1021 project number 197785619 (F.W., T.H., R.K.H.), GRK2355 project number 325443116 (F.W., L.S., P.B., A.G., O.R.), the Swedish Research Council 2018-05766 (F.W.), the Ministerio de Ciencia e Innovación/Agencia Española de Investigación MCIN/AEI (A.B.), and the Fondo Europeo de Desarrollo Regional (FEDER) by the European Union project PID2021-122567OB-I00 granted by MCIN/AEI /10.13039/501100011033 / FEDER, EU (A.B.).

## Author contributions

L.S., R.K.H., O.R., and F.W. designed the research. L.S., M.J.P., K.K., B.O., K.H., and O.R. acquired the data. P.B. processed the data. L.S., P.B., M.J.P., K.K., S.J., G.L., A.B., A.G., O.R., and F.W. analyzed and interpreted the data. L.S., A.B., A.G., T.H., R.K.H., O.R., and F.W. acquired the funding. L.S. and F.W. wrote the paper. All authors reviewed and approved the final manuscript.

## Funding

## Competing interests
