## [Peer Review File · Nature Communications]

Nucleocapsids of the Rift Valley fever virus ambisense S segment contain an exposed RNA element in the center that overlaps with the intergenic regionREVIEWER COMMENTS

Reviewer #1 (Remarks to the Author):

The manuscript entitled "Rift Valley fever virus nucleocapsids contain an exposed RNA sequence on the ambisense S segment that is not identical with the intergenic region", describes an interesting study in which the interactome of the Rift Valley fever virus nucleocapsid protein with all the three different viral genome segments was mapped. Using the iCLIP(2) technology not only a region in the S-segment was identified that is poorly covered by N protein, follow-up experiments revealed that this region is sensitive to RNase digestion and can be targeted by antisense oligonucleotides. Furthermore, RNA modelling suggests that an RNA loop might be responsible for the observed low N coverage and is in fact the functional unit essential for allowing ambisense transcription. Altogether this paper presents a unique dataset that provides a novel view on how some negative strand RNA viruses increase their genomic space. This being said, the manuscript could be improved by addressing the following points.

- 1) The authors show that the S segment contains a central non-encapsidated region but conclude this region has a functional role. Although a specific functional role for this region is highly likely, the authors did not experimentally zoomed into the functional characterization of this region. At a few places in the manuscript a bit more reservation on the functional characterization statements would be appropriate.
- 2) Compared to the rest of the manuscript the abstract is not optimally written and could benefit from editing. Line 10, add 'the'. What is the evidence for 'stable' stem loop structures. Could be flexible during certain stages of replication/transcription?
- 3) The title of the manuscript seems not optimal. The word "Identical" does not seem to fully match the message. Furthermore, the sentence subject should probably be the Ambisense S segment, now it reads RVFV nucleocapsids (which is not accurate). It reads that all nucleocapsids contain an exposed region, but it is only the S-segment.
- 4) Line 45, Does the accessibility relate to differences in RNA binding sites on monomeric N or by different 3D structures of the RNP, or both?
- 5) line 46: representative ...for other negative strand RNA viruses?.
- 6) In the introduction section very little is mentioned about the available crystal structure data of RVFV-N specifically, and the potential RNA binding sites in an N molecule. What was

the hypothesis upfront for RVFV RNP encapsidation?

7) Replication and especially encapsidation is expected to be initiated around 3-5 h post infection. The rationale for choosing the 5 hour time point, which seems very early is not fully clear. At such an early timepoint encapsidation of various RNA templates might not have reached its matured state. Have the authors also tested other (later) timepoints?

8) In the manuscript no attention is given to potential sequencing bias. As part of the iCLIP technology two PCR based amplification steps seem to be needed. Furthermore, various adapters have to be ligated. All of these (necessary) processing steps include sequencing bias and influence coverage, the end result. At a minimum this bias should be discussed.

9) Related to point 8, how do the authors explain the seemingly higher 3' coverage, especially in the MP-12 virion population?

10) Though the overall evidence for the presence of a stem loop structure in the S-segment is convincing, all the provided evidence is indirectly. Have the authors tried to obtain (direct) visual evidence of a stem loop structure in the S-segment by using electron microscopy techniques such as those presented here The Native Orthobunyavirus Ribonucleoprotein Possesses a Helical Architecture | mBio (asm.org)

11) Have the authors considered to assess the purified RNPs by LC/MSMS to check for the presence of the polymerase and for any host proteins attached to the RNPs? Besides N-RNA interactions also other protein-RNA interactions might be involved in the observed phenotypes.

12) In the discussion the authors state that their results are more or less in line with a selective genome packaging strategy, though the arguments for this conclusion are poor and seem primarily based on the indirect assessment of segment ratios using the indirect NGS approach which, as said before, is prone to sequencing bias. You could also argue that the iCLIP analyses in which no large exposed regions on the L or M segments and most of the S segments was observed (in contrast to Influenza) are fully in line with a random genome packaging strategy.

13) In the discussion section the obtained results should preferably also be reflected against reports showing an interaction of (oligomerized) N-protein with the cytoplasmic tails of the glycoproteins. Will segment variabilities in N coverage play a role in RNP packaging efficiencies?

14) In the discussion section discussion of potential follow up experiments such as the use

of site directed mutants could be valuable.

Reviewer #2 (Remarks to the Author):

This is a very interesting manuscript describing a variation on CLIP-seq (iCLIP) on Rift Valley fever virus (RVFV). This is an important virus causing significant disease in people and life stock in Africa. The authors start with a careful comparison and validation of their procedures using two different antibodies against the nucleoprotein (N) of the virus as well as multiple independent repeats. From their analysis, they identify a region on the small segment that appears to be free of N. Upon further investigation this region overlaps with the intergenic region (IGR) that separates the two ORFs on opposite sides of the gene-segment. Next, the authors confirm that this region is not bound by N using anti-sense oligonucleotide targeted RNase H digestions assays. This region is predicted to form a large hairpin with multiple double stranded sections. Finally, they used antisense oligonucleotides targeting this region to show that this part of the genome is important for virus replication, albeit the effects on virus replication and virus titers are very modest.

The iCLIP and validation of their results are well done and described in this manuscript. Their conclusions that there is a nucleoprotein free region in the small segment is supported by the data and additional experimentation. However, the functional analysis is not convincing and the effects of the antisense oligonucleotides are not necessarily linked to the exposed and double stranded RNA. To claim that this region is more important than other regions or that anti-sense oligo's can only target the exposed double stranded RNA, additional oligo targeting other segments or the small segment at the 5' or 3' end need to be included. The manuscript would also benefit from additional RNA probing techniques like DMS-MaPseq to confirm the existence of the structure in cells or virus. Finally, the paper would benefit from the analysis of mutant viruses with synonymous and structural changes to this region showing either the impact of these changes on replication or N binding. This reviewer would like to see one of these additional pieces of data added to the manuscript.

Additional comments:

1) The title of the manuscript is technically correct, but somewhat misleading. The N-free

region is larger than the IGR, but it does include it. Therefore I would suggest changing the title

2) The manuscript seems to go back and forth on whether the N-free (or N-low) region does or does not contain the IGR (title, lines 9-10, line 54, versus line 134 for example). And if these are truly different, and this reviewer is not convinced they are, why is that important? I think it is very exciting that IGR is part of a much larger functional RNA element in the small segment that extends into the coding regions of N and NSs.

3) How are the exposed regions defined? Statistical or eye-balling it? Looking at 7A, the EvSR and the RNase H sensitive region look similar to me. Why is ASO6 not included in your EvSR, but ASO3 is?

4) The 3' and 5' ends are over represented in the iCLIP experiments? Is this evidence of defective interfering particles or copy-back gene-segments?

Minor comments:

1) Can you comment on the safety level and biosafety measures for MP-12 and clone 13 in the manuscript?

2) Sometimes the method is called N-CLIP and other times it is iCLIP or iCLIP2

3) Why was AMO added again after the transfection and infection? What this with transfection reagents?

Reviewer #3 (Remarks to the Author):

The manuscript by Shalamova et al., is well-written and high-quality report of studies using iCLIP (individual-nucleotide resolution UV crosslinking and immunoprecipitation) to map interactions of the viral RNA (vRNA) and complementary RNA (cRNA) with the viral nucleocapsid (N) protein. The work provides evidence that part of the intergenic region (IGR) on the small (S) segment of Rift Valley fever virus (RVFV) is not encapsidated by the N protein. The authors confirm the mapping of exposed regions by ASO-targeted RNase cleavage and by targeting the exposed domains with antisense morpholino oligonucleotides. I value the fact that the authors used proper controls and repeats as well as independent additional methods to confirm their findings. Although the findings are not surprising (it is generally assumed that the S segment contains a stem-loop structure that is not associated

with N protein), the results are valid, significant, and important to the field.

Specific comments:

The manuscript states that “the prevailing view on negative-strand RNA viruses, including phleboviruses, is that the vRNAs and cRNAs are entirely and uniformly bound by N protein” (lines 38-40). As noted above, I do not agree with this suggestion. In contrast, it is generally believed that the IGR is not bound by N protein and that this region forms a stem-loop structure. Nevertheless, this was never reported in literature, making the study valuable to the field.

The title of the manuscript is not the strongest in my opinion, as it does not reflect the most important conclusion.

Lines 10-11: I suggest to not use EvSR and EcSR in the abstract, as these abbreviations are unclear at this position in the manuscript.

Line 13-14: “The RVFV ambisense S segment thus contains a central non-encapsidated RNA region with a functional role”. The study does not demonstrate such functional role.

Line 28: “...the cRNAs replicated back into vRNAs”. This sentence should be written in present tense. “replicated back” should be “replicate back”. However, the word “replicate” may not be the most appropriate.

Line 24: “whose” refers to a person.

Lines 72-74: This description is not completely clear. “The radioactive complexes were excised from the membrane at a size range that is expected for N and its multimers”. It would be good to specify that the multimer of N is a hexamer and what Mw range was excised.

Line 75: “at which a short peptide remained”. Please clarify this short peptide.

Fig. 1. Panel C contains some very small fonts. Please increase the size.

Fig. 6: I do not understand why the RNP is more sensitive to RNase as naked RNA at the concentration used to produce lane 4. Specifically, almost all RNP is degraded in lane 4, while a significant amount of naked RNA is present in the same conditions. Similar results are observed in panel B.

Line 200: I would not refer to this as “in vivo” as the work was done in cell culture (in vitro).

Line 254: “...are generated at all”. Suggest reformulating this (or remove “at all”).

Line 265: What is meant with “antiterminated”? Suggest using another term.

Lines 280-283: I do not agree that the surplus amounts of genomic M contradict the studies described in references 32 and 33. While the authors suggest they have analyzed virions, they in fact analyzed quite crude material that also contains (v/c)RNA released from the cells, while the studies referenced in 32 and 33 have visualized individual virions both inside and outside infected cells. The latter studies have demonstrated that surplus M genome segments (generally detected by PCR or sequencing) do not (only) represent virions. Also, the suggestion that predominance of vM makes sense because the M segment organizes the co-packaging of L and S segments is not appropriate since most evidence suggests that this hypothesis is incorrect. A two-segmented RVFV variant lacking the M-segment has been described (Brennan et al., J Virol 85: 10310–10318) and replicon particles containing only S and L genome segments were produced very efficiently (Kortekaas et al., J Virol 85: 12622–12630; Dodd et al., J Virol 86:4204–4212). Clearly, the M segment does not “organize the co-packaging of L and S segments”.

Reply to the reviewers' comments

Reviewer 1:

The manuscript entitled "Rift Valley fever virus nucleocapsids contain an exposed RNA sequence on the ambisense S segment that is not identical with the intergenic region", describes an interesting study in which the interactome of the Rift Valley fever virus nucleocapsid protein with all the three different viral genome segments was mapped. Using the iCLIP(2) technology not only a region in the S-segment was identified that is poorly covered by N protein, follow-up experiments revealed that this region is sensitive to RNase digestion and can be targeted by antisense oligonucleotides. Furthermore, RNA modelling suggests that an RNA loop might be responsible for the observed low N coverage and is in fact the functional unit essential for allowing ambisense transcription. Altogether this paper presents a unique dataset that provides a novel view on how some negative strand RNA viruses increase their genomic space.

Reply:

We thank the reviewer for the very positive and encouraging comments on our manuscript and data.

This being said, the manuscript could be improved by addressing the following points.

1) The authors show that the S segment contains a central non-encapsidated region but conclude this region has a functional role. Although a specific functional role for this region is highly likely, the authors did not experimentally zoomed into the functional characterization of this region. At a few places in the manuscript a bit more reservation on the functional characterization statements would be appropriate.

Reply:

We agree that our interpretations on a functional role of the central non-encapsidated region (EvSR) should

have been toned down in the previous manuscript. In response to this comment and also to similar comments by reviewers 2 and 3, we have added new experiments in which we inserted the EvSR in an S segment reporter construct. We observed that replacing the conventional IGR with the EvSR indeed boosted reporter expression, virus-like particle production and replication of a recombinant virus, and that mutating the nucleotides involved in the predicted stem weakens the positive effect of the EvSR. Based on these new data (shown as new figure 9), we are confident that the EvSR indeed has a functional role, so we dared to leave this conclusion in the abstract.

2) Compared to the rest of the manuscript the abstract is not optimally written and could benefit from editing. Line 10, add 'the'. What is the evidence for 'stable' stem loop structures. Could be flexible during certain stages of replication/transcription?

Reply:

We now worked more on the abstract. The idea to call the stem loop structure "stable" was due to the low minimum free energy calculated by the structure prediction program, but we agree that we should not emphasize this too much, so we took it out. Moreover we are happy to integrate the idea with the flexible structure into the discussion (line 339).

3) The title of the manuscript seems not optimal. The word "Identical" does not seem to fully match the message. Furthermore, the sentence subject should probably be the Ambisense S segment, now it reads RVFV nucleocapsids (which is not accurate). It reads that all nucleocapsids contain an exposed region, but it is only the S-segment.

Reply:

Thank you for the really helpful comment, which was also made by experts 2 and 3. We have changed the title to "Nucleocapsids of the Rift Valley fever virus ambisense S segment contain an exposed RNA sequence in the center that overlaps with the intergenic region"

4) Line 45, Does the accessibility relate to differences in RNA binding sites on monomeric N or by different 3D structures of the RNP, or both?

Reply:

Influenza virus NP has a positively charged RNA binding groove, but the RNA is wound around each NP (hence the unusually big number of 24 bases per NP) rather than buried inside the RNPs as in other segmented NSVs. We have modified the sentence to clarify this (lines 46 to 51).

5) line 46: representative ...for other negative strand RNA viruses?.

Reply:

According to the literature (Sabsay et al. 2023, Ref. 6), there also are some non-segmented NSVs which have their vRNA somewhat exposed on the outside (whereas others have it hidden), but none of them as much as influenza virus has. But since for the non-segmented NSVs this seems not be entirely clear, we now restricted our statements on the RNA location within the RNPs to the segmented NSVs only (see line 53).

6) In the introduction section very little is mentioned about the available crystal structure data of RVFV-N specifically, and the potential RNA binding sites in an N molecule. What was the hypothesis upfront for

RVFV RNP encapsidation?

Reply:

As wished, we now added a sequence on the structure-derived knowledge how RVFV N binds and encapsidates RNA (lines 49 to 51).

7) Replication and especially encapsidation is expected to be initiated around 3-5 h post infection. The rationale for choosing the 5 hour time point, which seems very early is not fully clear. At such an early timepoint encapsidation of various RNA templates might not have reached its matured state. Have the authors also tested other (later) timepoints?

Reply:

The cellular RNPs we had analysed at the 5 h time point because we wanted to ensure that cellular resources are plentiful, replication just got in full swing, but RNPs are not yet exported into particles.

Inclusion of fully matured RNPs was ensured by the parallel analysis of RNPs that had been packaged into virions. We now explain this briefly in lines 94 to 95.

8) In the manuscript no attention is given to potential sequencing bias. As part of the iCLIP technology two PCR based amplification steps seem to be needed. Furthermore, various adapters have to be ligated. All of these (necessary) processing steps include sequencing bias and influence coverage, the end result. At a minimum this bias should be discussed.

Reply:

Since the CLIP technology was first introduced in 2003, there have been several approaches to clarify a potential sequence bias. These findings were summarized in PMID 28093074 (Ref 22). There are indeed biases both at starts and ends of CLIP reads for various reasons. The group of Jernej Ule (who had developed iCLIP back in 2003) found that having a diversity of read lengths in sequencing can decrease the impact of such read biases. We have used a read length of 150 nt, which includes a wide range of PCR product sizes. We are therefore confident that any sequence biases in our data are not systematically distorting the results, since the crosslink site has varying distances to the other end of the cDNA.

We now briefly discuss this in lines 137 to 139.

9) Related to point 8, how do the authors explain the seemingly higher 3' coverage, especially in the MP-12 virion population?

Reply:

This was also asked by reviewer 2, but we honestly don't know. If it were from DI particles, we would expect higher coverage at both the 5' and the 3' end (conventional DIs) or at the 5' end of the genome and the 3' end of the antigenome (copy-back DIs), but not preferentially on the 3' end that is actually synthesized last by the viral RdRP. Moreover, if it were DIs we would have expected to see the same overrepresentation of the ends in infected cells (MOI was 1), which is however not the case. BTW we prepared the virus stocks by using an MOI of 0.0005 to avoid DI formation. An explanation we favour more is that the higher encapsidation of the 3' ends are either a replication start signal (in case the RNP is the template) or a replication stop signal (in case the RNP is the product), but at this stage we can only speculate. BTW also Lee

et al. (Ref. 15) saw increased NP binding at either the 5' or the 3' ends of influenza virus segments. We now mentioned the observation and our thoughts about it in lines 111 to 123.

- 0) Though the overall evidence for the presence of a stem loop structure in the S segment is convincing, all the provided evidence is indirect. Have the authors tried to obtain (direct) visual evidence of a stem loop structure in the S segment by using electron microscopy techniques such as those presented here: The Native Orthobunyavirus Ribonucleoprotein Possesses a Helical Architecture | mBio (asm.org)

Reply:

The microscopy methods used in said paper allowed to determine the overall RNP architecture of Bunyamwera virus but would not be optimal for visualising naked RNA. However, we do plan to determine the EvSR 3D structure by SHAPE or related methods in future experiments. Since we need to do this in the native (RNP) context, it will be a major endeavour which we feel is beyond the scope of our iCLIP2 study. So far this has only been done for influenza virus particles, as described in two extensive studies (Dadonaitė et al., Nat Microbiology 2019 and Mirska et al., CMLS 2023; Refs 17 and 18).

- 1) Have the authors considered to assess the purified RNPs by LC/MS/MS to check for the presence of the polymerase and for any host proteins attached to the RNPs? Besides N-RNA interactions also other protein-RNA interactions might be involved in the observed phenotypes.

Reply:

We had indeed started with Mass Spec analyses of immunoprecipitated RVFV RNPs. The L protein was co-precipitated as expected, and in addition we also observed an interaction with the cellular protein DDX17. From these results we cannot say which one of the segments interacts with DDX17, but it was previously shown to attach to RVFV M and S RNPs (Moy et al., Ref 35). So our so far unpublished data confirms this work which we had mentioned in our discussion (lines 341 to 343).

As our Mass Spec data need to be repeated (and are not entirely in the scope of our current work) and don't tell per se where those proteins are binding to the RNPs, we would prefer to not make them part of this manuscript. But as said the DDX17 literature is discussed and we will follow this up.

- 2) In the discussion the authors state that their results are more or less in line with a selective genome packaging strategy, though the arguments for this conclusion are poor and seem primarily based on the indirect assessment of segment ratios using the indirect NGS approach which, as said before, is prone to sequencing bias. You could also argue that the iCLIP analysis in which no large exposed regions on the L or M segments and most of the S segments was observed (in contrast to Influenza) are fully in line with a random genome packaging strategy.

Reply:

This point was also brought up by reviewer 2, and we agree with the comments. During the course of the manuscript revision, we had initially re-written the part with the RNA ratios. However, later on, when we reorganized the paper and had to integrate new results, we decided to free manuscript space by entirely removing the segment quantifications from the results and the discussion. The RNA ratios in the particles were a mere side result and their discussion needs a lot of space due to the mentioned controversies in the field. It is also not really within the scope of our paper and their removal helps to streamline the story.

- 3) In the discussion section the obtained results should preferably also be reflected against reports showing

an interaction of (oligomerized) N-protein with the cytoplasmic tails of the glycoproteins. Will segment variabilities in N coverage play a role in RNP packaging efficiencies?

Reply:

This is a very good point but as written above, we had to remove the RNA ratio part and the discussion on packaging entirely to focus on the EvSR as the key result of our study. So it is not necessary to discuss this point.

14) In the discussion section discussion of potential follow up experiments such as the use of site directed mutants could be valuable.

Reply:

In fact, we now generated and mutated virus-like particles to exactly address this, as described in our reply to point 1.

Reviewer #2:

This is a very interesting manuscript describing a variation on CLIP-seq (iCLIP) on Rift Valley fever virus (RVFV). This is an important virus causing significant disease in people and life stock in Africa. The authors start with a careful comparison and validation of their procedures using two different antibodies against the nucleoprotein (N) of the virus as well as multiple independent repeats. From their analysis, they identify a region on the small segment that appears to be free of N. Upon further investigation this region overlaps with the intergenic region (IGR) that separates the two ORFs on opposite sides of the gene segment. Next, the authors confirm that this region is not bound by N using anti-sense oligonucleotide targeted RNase H digestion assays. This region is predicted to form a large hairpin with multiple double stranded sections. Finally, they used antisense oligonucleotides targeting this region to show that this part of the genome is important for virus replication, albeit the effects on virus replication and virus titers are very modest.

Reply:

We are very grateful for these positive and encouraging comments on our manuscript and data.

The iCLIP and validation of their results are well done and described in this manuscript. Their conclusions that there is a nucleoprotein free region in the small segment is supported by the data and additional experimentation. However, the functional analysis is not convincing and the effects of the antisense oligonucleotides are not necessarily linked to the exposed and double stranded RNA. To claim that this region is more important than other regions or that anti-sense oligos can only target the exposed double stranded RNA, additional oligo targeting other segments or the small segment at the 5' or 3' end need to be included.

Reply:

We agree, the experiments regarding inhibitory effects of the RNA antisense morpholino oligonucleotides (AMOs) targeting different regions on the S segment (former figure 9) would need improvement. We had ordered further AMOs that also included N-covered S segment target regions. However, during the course of these experiments we had to realize that one of the AMOs that was from the same supplier as before, Gene Tools LLC, led to contaminations of our cell culture. As this made any interpretation of the – as the reviewer correctly remarked – anyway quite modest effects of the AMOs basically impossible, we had to drop this approach for testing the functional importance of the exposed region.

As an alternative (and also following the advice of all three reviewers), we inserted the EvSR into a RVFV reporter S segment and measured its effect on S segment transcription/replication, packaging into virus-like particles, and also on the multiplication of a recombinant virus. Compared to the canonical IGR, all those activities were higher when the S segment contained the full EvSR. Moreover, for the VLPs we also mutated the EvSR in a way that weakens the base pairings and again found that wild-type EvSR performed best (mutated viruses could not be rescued). The results from these experiments, replacing the AMO data of figure 9 and described in a new chapter (lines 232 to 285) clearly show that the EvSR boosts S segment activity, indicating a functional relevance.

The manuscript would also benefit from additional RNA probing techniques like DMS-MaPseq to confirm the existence of the structure in cells or virus.

Reply:

Mapping the structure in vivo is the next goal we have. However, the proposed DMS-MaPseq is time-consuming to establish, requires us to look out for new collaborators, and would in our view be an entire new paper, comparable to the ones published for influenza viruses (Refs 17 and 18). We hope that our new data on the functional importance of the EvSR, the first report on such an RNA element for any ambisense virus, is sufficient to complete our study.

Finally, the paper would benefit from the analysis of mutant viruses with synonymous and structural changes to this region showing either the impact of these changes on replication or N binding. This reviewer would like to see one of these additional pieces of data added to the manuscript.

Reply:

As mentioned above, we had done this now. The VLPs and recombinant viruses we had generated show that the EvSR is superior to the IGR in supporting RVFV transcription and replication. Moreover, we mutated the EvSR in the VLP system and found that weakening the predicted dsRNA regions lowered the EvSR-dependent activity. EvSR-mutated viruses could so far not be rescued.

Additional comments:

1) The title of the manuscript is technically correct, but somewhat misleading. The N-free region is larger than the IGR, but it does include it. Therefore I would suggest changing the title

Reply:

Also reviewers 1 and 3 found the title sub-optimal, so we changed it to “Nucleocapsids of the Rift Valley fever virus ambisense S segment contain an exposed RNA element in the center that overlaps with the intergenic region”

2) The manuscript seems to go back and forth on whether the N-free (or N-low) region does or does not contain the IGR (title, lines 9-10, line 54, versus line 134 for example). And if these are truly different, and this reviewer is not convinced they are, why is that important? I think it is very exciting that IGR is part of a much larger functional RNA element in the small segment that extends into the coding regions of N and NSs.

Reply:

Yes, our most important finding is that there is an exposed RNA element in the S segment that includes the IGR but also adjacent coding regions. For virion RNPs, the EvSR indeed includes the entire IGR (iCLIP, RNase H protection, see Fig. 6). iCLIP data from infected cells however showed an N peak in the “left” half of the IGR which may mean that at least some RNPs are not exposing the entire region (Fig. 4). This is even more obvious for Clone 13 (Fig. 9), where high N peaks are starting an encapsidated zone from the middle of the IGR on towards its 3’ end.

We therefore cautiously defined the shorter version, which is consensus between the different methods and RNP populations, as the “core” EvSR. Of note, the predicted structures of the core EvSR and the longer “RNaseH” version are not substantially different, the longer sequence just has additional structures at its bottom (see Fig. S6). The definition of the EvSR is now given in lines 172 to 174.

3) How are the exposed regions defined? Statistical or eye-balling it? Looking at 7A, the EvSR and the RNase H sensitive region look similar to me. Why is ASO6 not included in your EvSR, but ASO3 is?

Reply:

The RNaseH sensitive region (now Fig. 6) consists of the core EvSR plus adjacent sequences on both sides. We conservatively define the core EvSR as the unencapsidated region between the (in virions comparatively low) N peaks at positions 800 and 900, respectively. The long EvSR is defined by RNaseH protection. ASO6 is situated 3’ of the 3’ N peak of the core EvSR, whereas ASO3 has the 5’ N peak in its middle. The left hand part of the ASO3 sequence (Fig. 7A) is therefore part of the core EvSR, whereas ASO6 isn’t. As mentioned, the definitions are now explained in line 172 ff.

4) The 3’ and 5’ ends are over represented in the iCLIP experiments? Is this evidence of defective interfering particles or copy-back gene-segments?

Reply:

This was also noted by reviewer 1. We actually don’t think it is from conventional or copy-back DI particles with internal deletions, as most of the peaks were at the 3’ ends. For conventional DIs with internal deletions we would have expected higher coverage at both the 5’ and the 3’ end, and for copy-back Dis a higher coverage of the genome 5’ end and of the antigenome 3’ends, and not preferentially on the 3’ end that is actually synthesized last by the viral RdRP. Moreover, if it were DIs we would have expected to see the same overrepresentation of the ends in infected cells, which is not the case. BTW we prepared the virus stocks by using an MOI of 0.0005 to avoid DI formation. One possibility could be that the higher encapsidation densities at the 3’ ends either serve as a replication start signal (in case the RNP is the template) or a replication stop signal (in case the RNP is the product). But this is mere speculation. In line with our data also Lee et al. (Ref. 15) saw increased NP binding at either the 5’ or the 3’ ends of influenza virus segments.

We now mentioned the observation and our thoughts on it in lines 114 to 124.

Minor comments:

1) Can you comment on the safety level and biosafety measures for MP-12 and clone 13 in the manuscript?

Reply:

Thanks for pointing at this omission from our side. Both strains are BSL-2, and we now included this information in the Materials and Methods (line 387).

2) Sometimes the method is called N-CLIP and other times it is iCLIP or iCLIP2

Reply:

Thanks. We now uniformly call the method iCLIP2. N-CLIP (now N-iCLIP2) is used to emphasize that we pulled down the nucleoprotein, and we would like to keep this if the reviewer agrees.

3) Why was AMO added again after the transfection and infection? What this with transfection reagents?

Reply:

This was to keep the AMOs at levels high enough to interfere with the replicating virus (yes, with transfection agent). As remarked by this reviewer, even then the inhibitory effects were modest, which was one reason why we related the AMO experiments with the much more telling reverse genetics data.

Reviewer #3:

The manuscript by Shalamova et al., is well-written and high-quality report of studies using iCLIP (individual-nucleotide resolution UV crosslinking and immunoprecipitation) to map interactions of the viral RNA (vRNA) and complementary RNA (cRNA) with the viral nucleocapsid (N) protein. The work provides evidence that part of the intergenic region (IGR) on the small (S) segment of Rift Valley fever virus (RVFV) is not encapsidated by the N protein. The authors confirm the mapping of exposed regions by ASO-targeted RNase cleavage and by targeting the exposed domains with antisense morpholino oligonucleotides. I value the fact that the authors used proper controls and repeats as well as independent additional methods to confirm their findings. Although the findings are not surprising (it is generally assumed that the S segment contains a stem-loop structure that is not associated with N protein), the results are valid, significant, and important to the field.

Reply:

Thank you! We are very pleased by these positive and encouraging comments.

Specific comments:

The manuscript states that “the prevailing view on negative-strand RNA viruses, including phleboviruses, is that the vRNAs and cRNAs are entirely and uniformly bound by N protein” (lines 38-40). As noted above, I do not agree with this suggestion. In contrast, it is generally believed that the IGR is not bound by N protein and that this region forms a stem-loop structure. Nevertheless, this was never reported in literature, making the study valuable to the field.

Reply:

It is indeed assumed by the community (and we found one paper where this was mentioned) that the IGR must be exposed in some way and form a secondary structure. We now removed the phlebovirus mention in that sentence and added this sentence in the introduction of the IGR: “The IGR contains the transcription termination motifs of the two reading frames and was suspected to fold into a stem-loop (Ref.8)” (line 37).

The title of the manuscript is not the strongest in my opinion, as it does not reflect the most important conclusion.

Reply:

Also reviewers 1 and 2 have noted this, so we changed it to “Nucleocapsids of the Rift Valley fever virus ambisense S segment contain an exposed RNA element in the center that overlaps with the intergenic region”

Lines 10-11: I suggest to not use EvSR and EcSR in the abstract, as these abbreviations are unclear at this position in the manuscript.

Reply:

True, we removed the acronyms from the abstract.

Line 13-14: “The RVFV ambisense S segment thus contains a central non-encapsidated RNA region with a functional role”. The study does not demonstrate such functional role.

Reply:

This comment is in line with the ones by reviewers 1 and 2 who found the results from the AMO experiments - done to investigate functional importance of the EvSR - weak.

In order to address these comments, we now generated recombinant viruses and virus-like particles (VLP) expressing a reporter gene on an S segment containing either only the IGR or the entire EvSR. The results of those experiments (new figure 9) clearly show that the wild-type EvSR is required for the optimal S segment function. In the light of these new results, we would prefer to leave that conclusion in the abstract.

Line 28: “...the cRNAs replicated back into vRNAs”. This sentence should be written in present tense. “replicated back” should be “replicate back”. However, the word “replicate” may not be the most appropriate.

Reply:

Thank you, the “replicated” was actually not intended as being past tense, but as present passive voice form connected to the half sentence before. As the “are” is quite far away in the sentence, we changed to “Upon cell infection, the vRNA segments are individually transcribed into mRNAs (primary transcription), subsequently replicated into antigenomes (cRNAs), and the cRNAs are then replicated back into vRNAs.” (lines 26 to 28)

Line 24: “whose” refers to a person.

Reply:

We changed the sentence to “Yet, phleboviruses like RVFV have expanded their genetic space by placing an additional reading frame (for the nonstructural protein NSs) on the S segment which is transcribed from the cRNA” (lines 32 to 34).

Lines 72-74: This description is not completely clear. “The radioactive complexes were excised from the membrane at a size range that is expected for N and its multimers”. It would be good to specify that the multimer of N is a hexamer and what Mw range was excised.

Reply:

To make this sentence comprehensible, we changed it to “The radioactive complexes were excised from the membrane at a size range of 57 to 230 kDa that is expected for N (approximately 27 kDa) and its hexamers

(Ref 9) plus 20 to 60 kDa of RNA and adapter” (lines 79 to 81).

Line 75: “at which a short peptide remained”. Please clarify this short peptide.

Reply:

After proteinase K cleavage, there remains at least one amino acid or a small peptide at the crosslink side. We have added this information in the text (lines 82 to 83).

Fig. 1. Panel C contains some very small fonts. Please increase the size.

Reply:

Thank you, we have increased the size of the smallest fonts in Figure 1C

Fig. 6: I do not understand why the RNP is more sensitive to RNase as naked RNA at the concentration used to produce lane 4. Specifically, almost all RNP is degraded in lane 4, while a significant amount of naked RNA is present in the same conditions. Similar results are observed in panel B.

Reply:

We explain this by considering that RNPs expose only one RNase-sensitive site, namely the unencapsidated region in the middle of the S segment, whereas naked RNAs have potential RNA cleavage sites over their entire length. If at intermediate RNase concentrations there would be on average 1 cleavage per segment, it would always be in the same region in case of the RNPs, but randomly distributed in case of the naked RNAs. Hence, where there is a distinct band for the RNPs, there is a slight increase in smear for the naked RNAs.

Line 200: I would not refer to this as “in vivo” as the work was done in cell culture (in vitro).

Reply:

True. We have removed this, as the data (former figure 9) on inhibitory antisense oligonucleotides are now replaced by the much more telling reverse genetics experiments in new figure 9.

Line 254:”...are generated at all”. Suggest reformulating this (or remove “at all”).

Reply:

We have removed the “at all” as advised.

Line 265: What is meant with “antiterminated”? Suggest using another term.

Reply:

Agreed, that needs an explanation. Antitermination refers to a regulatory mechanism that allows RNA polymerases to bypass transcription termination signals, thus enabling continued transcription elongation. We have changed the sentence as follows: “it is unclear how an exposed region in the middle of a nucleocapsid is regulated to either terminate mRNA transcription, or bypass the mRNA termination signal for genome replication” (lines 333 to 335).

Lines 280-283: I do not agree that the surplus amounts of genomic M contradict the studies described in

references 32 and 33. While the authors suggest they have analyzed virions, they in fact analyzed quite crude material that also contains (v/c)RNA released from the cells, while the studies referenced in 32 and 33 have visualized individual virions both inside and outside infected cells. The latter studies have demonstrated that surplus M genome segments (generally detected by PCR or sequencing) do not (only) represent virions. Also, the suggestion that predominance of vM makes sense because the M segment organizes the co-packaging of L and S segments is not appropriate since most evidence suggests that this hypothesis is incorrect. A two-segmented RVFV variant lacking the M-segment has been described (Brennan et al., J Virol 85: 10310–10318) and replicon particles containing only S and L genome segments were produced very efficiently (Kortekaas et al., J Virol 85: 12622–12630; Dodd et al., J Virol 86:4204–4212). Clearly, the M segment does not “organize the co-packaging of L and S segments”.

Reply:

We agree with this criticism, which was also issued by reviewers 1 and 2. As we had to free more manuscript space for our reverse genetics data (as well as due to the fact that the segment ratios are not the focus of our study), we removed both the data and our criticised interpretation from the manuscript.

Everybody writes this in their rebuttal, but I really mean it: We are very grateful to all three reviewers for helping us to strengthen and streamline our manuscript.

We hope that it can now be considered suitable for publication in *Nature Communications*.

REVIEWERS' COMMENTS

Reviewer #1 (Remarks to the Author):

The authors appropriately addressed my comments.

Reviewer #2 (Remarks to the Author):

The authors addressed all the reviewers comments and added some interesting and important additional information with the reporter assay. I have no further comments.

Reviewer #3 (Remarks to the Author):

I thank the authors for addressing most of my points appropriately. However, the newly added data and conclusions have raised some concern. The authors now acknowledge that it was already described in literature that the S segment is not completely covered by N protein. They refer to reference 8, and state that the “IGR contains the transcription termination motifs of the two reading frames and was suspected to form into a stem-loop structure”. However, the paper that they refer to actually does not suggest the formation of a stem-loop. In contrast, the paper states the following: “Extensive RNA secondary structure prediction analysis of the S segment intergenic regions and M segment UTRs of these viruses failed to predict convincing high-energy hairpin structures which could be involved in the transcription termination process”. The authors of this earlier work also performed experiments using reverse genetics, and actually concluded “These data showed that the N and NSs termination signals function independently and do not form stem-loop structures”. The same authors also evaluated putative hairpin structures for Punta Toro and Uukuniemi virus and concluded that “these would be of such low energy or high complexity as to be unlikely to form”. Finally, the same authors also reported that they were able to “rescue recombinant viruses that failed to correctly terminate their N or NSs mRNA due to deletion of the respective transcription termination signal”. These viruses grew less efficiently, similar as described in the present paper.

Reply to the comment by reviewer #3

Reviewer #3:

I thank the authors for addressing most of my points appropriately. However, the newly added data and conclusions have raised some concern. The authors now acknowledge that it was already described in literature that the S segment is not completely covered by N protein. They refer to reference 8, and state that the “IGR contains the transcription termination motifs of the two reading frames and was suspected to form into a stem-loop structure”. However, the paper that they refer to actually does not suggest the formation of a stem-loop. In contrast, the paper states the following: “Extensive RNA secondary structure prediction analysis of the S segment intergenic regions and M segment UTRs of these viruses failed to predict convincing high-energy hairpin structures which could be involved in the transcription termination process”. The authors of this earlier work also performed experiments using reverse genetics, and actually concluded “These data showed that the N and NSs termination signals function independently and do not form stem-loop structures”. The same authors also evaluated putative hairpin structures for Punta Toro and Uukuniemi virus and concluded that “these would be of such low energy or high complexity as to be unlikely to form”. Finally, the same authors also reported that they were able to “rescue recombinant viruses that failed to correctly terminate their N or NSs mRNA due to deletion of the respective transcription termination signal”. These viruses grew less efficiently, similar as described in the present paper.

Reply:

Thank you for pointing this out, our statement that reference 8 suspected a stem-loop structure is indeed phrased in an unfortunate manner. Our sentence in the introduction was intended to accommodate the original comment by reviewer #3 in which he/she wrote that “it is generally believed that the IGR is not bound by N protein and that this region forms a stem-loop structure. Nevertheless, this was never reported in literature”. So to address this reviewer comment we were in the situation that on one hand we had to mention (and hence cite) a belief (in a stem-loop) that on the other hand is not published anywhere. However, there is this one paper (reference 8) in which the possibility of an IGR stem-loop is indeed discussed, and the authors cited several papers that suspected such a secondary structure using computer predictions. Here are the relevant quotes from reference 8:

- “Secondary structure prediction analysis suggested that this intervening intergenic region may be capable of forming hairpin structures, although less convincingly than for those found in arenaviruses” (refers to a paper on Punta Toro phlebovirus)
- “It was predicted that a short AU-rich hairpin may form within this region, although this potential structure was of relatively low energy” (refers to a paper on Uukuniemi phlebovirus)
-

From these literature references the authors of reference 8 concluded that “Such data...led to the prevailing view that mRNA transcription termination of ambisense genome RNA segments of bunyaviruses likely involves secondary structure mechanisms similar to those described for arenaviruses”.

So here we had the sought-after reference to accommodate the reviewer #3 comment about the “belief” of the field in a stem-loop structure. However, when the authors of reference 8 did their own structure predictions for RVFV and two other phleboviruses, they came to the conclusion that “However, extensive

analysis using secondary structure prediction algorithms failed to identify any likely high-energy structures for RVF, SFS, or TOS virus (data not shown)". There are also other text passages with similar statements, as cited by the reviewer #3.

In other words, reference 8 contains text passages on both opinions, namely the potential existence of a (low-energy) stem-loop in the IGR (citations of the Punta Toro and Uukuiniemi papers) as well as its the non-existence (own data by the ref 8 authors that was however not shown), and this is why we wrote that "it was suspected to fold into a stem-loop (8)". We agree however that "suspected" may be too one-sided and hence changed the sentence to "The IGR contains the transcription termination motifs of the two reading frames and was discussed to fold into a stem-loop, although the evidence is weak (8)" (lines 37 and 38).

Of note: the entire discussion here is about the IGR and not about the EvSR which we discovered in the present paper.

We hope that this text modification addresses the reviewer #3 comment in an appropriate manner, such that the manuscript can now be considered suitable for publication in *Nature Communications*.